# Conserved human effector Treg cell transcriptomic and epigenetic signature in arthritic joint inflammation

Gerdien Mijnheer[1,6], Lisanne Lutter [1,6], Michal Mokry [1,2,3], Marlot van der Wal [1], Rianne Scholman[1], Veerle Fleskens[4], Aridaman Pandit [1], Weiyang Tao[1], Mark Wekking[3], Stephin Vervoort[1,5], Ceri Roberts [4], Alessandra Petrelli[1], Janneke G. C. Peeters [1], Marthe Knijff[1], Sytze de Roock[1], Sebastiaan Vastert[1], Leonie S. Taams [4], Jorg van Loosdregt[1,5,7] & Femke van Wijk [1,7 ✉]

Treg cells are critical regulators of immune homeostasis, and environment-driven Treg cell differentiation into effector (e)Treg cells is crucial for optimal functioning. However, human Treg cell programming in inflammation is unclear. Here, we combine transcriptional and epigenetic profiling to identify a human eTreg cell signature. Inflammation-derived functional Treg cells have a transcriptional profile characterized by upregulation of both a core Treg cell (FOXP3, CTLA4, TIGIT) and effector program (GITR, BLIMP-1, BATF). We identify a specific human eTreg cell signature that includes the vitamin D receptor (VDR) as a predicted regulator in eTreg cell differentiation. H3K27ac/H3K4me1 occupancy indicates an altered (super-)enhancer landscape, including enrichment of the VDR and BATF binding motifs. The Treg cell profile has striking overlap with tumor-infiltrating Treg cells. Our data demonstrate that human inflammation-derived Treg cells acquire a conserved and specific eTreg cell profile guided by epigenetic changes, and fine-tuned by environment-specific adaptations.

[1] Center for Translational Immunology, Pediatric Immunology & Rheumatology, Wilhelmina Children's Hospital, University Medical Center Utrecht, Utrecht University, Utrecht, The Netherlands. [2] Regenerative Medicine Center Utrecht, Department of Pediatrics, University Medical Center Utrecht, Utrecht, The Netherlands. [3] Epigenomics facility, University Medical Center Utrecht, Utrecht, The Netherlands. [4] Centre for Inflammation Biology and Cancer Immunology, School of Immunology & Microbial Sciences, King's College London, London, UK. [5] Regenerative Medicine Center Utrecht, Center for Molecular Medicine, University Medical Center Utrecht, Utrecht, The Netherlands. [6] These authors contributed equally: Gerdien Mijnheer, Lisanne Lutter. [7] These authors jointly supervised this work: Jorg van Loosdregt, Femke van Wijk. ✉email: f.vanwijk@umcutrecht.nl

Forkhead box P3-expressing (FOXP3[+]) Treg cells are key players to control aberrant immune responses. Mutations in the *FOXP3* gene lead to severe autoimmunity and inflammation in both mice and humans[1,2]. Because of their potential for clinical applications, Treg cells have been intensively studied over the last decades.

There is accumulating evidence from mouse models that specific environments such as inflammation or non-lymphoid tissues can induce further differentiation/adaptation into specialized activated Treg cell subsets, also referred to as effector (e)Treg cells[3,4] (reviewed in ref. [5]). Characteristic for eTreg cells appears to be the maintenance of FOXP3 expression, increased expression of several molecules related to their function such as ICOS, CTLA4, and TIGIT, and adaptation to the local environment. For example, recent studies in mice show that in visceral adipose tissue Treg cells play a dominant role in metabolic homeostasis[6], whereas Treg cells in skin and gut can promote wound repair and are involved in local stem cell maintenance[7,8].

In mice, eTreg cells have been demonstrated to be crucial in specific inflammatory settings. Interestingly, inflammatory signals can induce the stable upregulation of typical Th cell transcription factors such as T-bet, allowing the Treg cells to migrate to the site of inflammation[9]. Moreover, co-expression of T-bet and Foxp3 is essential to prevent severe Th1 autoimmunity[10]. Studies using transgenic mice have contributed to the knowledge in this field, but how this can be translated to humans remains to be elucidated. Because of the interest in the use of Treg cells for therapeutic purposes in a variety of human diseases it is relevant to gain insight in human eTreg cell programming in inflammatory settings.

Also in humans, there is evidence of environment-induced adaptation of human (e)Treg cells[4,7,11] (reviewed in ref. [12]) including tumor-infiltrating Treg cell signatures[13–15]. Pertinent issues that remain to be addressed are how diverse and stable human eTreg cell programming is in the inflammatory environment, and whether this is regulated at an epigenetic level.

In this work, we investigate the gene expression profile and active enhancer landscape of human Treg cells in a human autoimmune-associated inflammatory environment. We show that in an inflammatory setting Treg cells differentiate into eTreg and become adapted to the environment both on a transcriptomic and epigenetic level. Furthermore, we demonstrate that there is substantial overlap between the synovial fluid (SF) Treg cell signature and recently published human tumor-infiltrating Treg cell signatures. These findings indicate that human eTreg cell programming is epigenetically imprinted and may be commonly induced in inflammatory conditions ranging from autoimmune to tumor settings.

## Results

**Enhanced core Treg cell signature in inflamed joints**. To investigate whether the gene expression profile of human Treg cells in an inflammatory environment is different from circulating human Treg cells, CD3[+]CD4[+]CD25[+]CD127[low] Treg cells and CD3[+]CD4[+]CD25[−]CD127[+] non-Treg cells were isolated from: synovial exudate obtained from inflamed joints of juvenile idiopathic arthritis (JIA) patients, peripheral blood (PB) from JIA patients with active and inactive disease, and PB from healthy children and healthy adults (Supplementary Fig. 1a for gating strategy). More than 90% of the sorted Treg cell populations were FOXP3 positive (Supplementary Fig. 1b). The transcriptional landscape was determined with RNA-sequencing. An unsupervised principal component analysis (PCA) was performed to study the variability between Treg cells derived from different environments. SF-derived Treg cells clearly clustered separately from PB-derived Treg cells (Fig. 1a), indicating that SF-derived

Treg cells have a specific expression pattern compared to PB-derived Treg cells.

In accordance with previous publications we confirmed that SF-derived Treg cells were functional using an in vitro suppression assay (Fig. 1b and Supplementary Fig. 2a)[16,17]. As expected, Treg cell signature genes were significantly enriched in SF Treg cells compared to SF CD4[+] non-Treg cells (Fig. 1c, left panel). Treg cell signature genes were also enriched in SF compared to PB Treg cells derived from both healthy children and JIA patients (Fig. 1c, right panel and Supplementary Fig. 2b). These included Treg cell hallmark genes important for Treg cell stability and function such as *FOXP3*, *CTLA4,* and *TIGIT*, at both the transcriptional and protein level (Fig. 1d, e and Supplementary Figs. 1a and 2c)[18,19]. In humans, CD3[+]CD4[+] non-Treg cells can upregulate FOXP3 and associated markers that as such can serve as markers for T cell activation[3]. We indeed observed a slight upregulation of FOXP3, CTLA4, and TIGIT at mRNA and protein level in SF non-Treg cells but not near the levels observed in SF Treg cells (Fig. 1d, e and Supplementary Figs. 1b and 2c), further confirming that SF Treg cells and non-Treg cells are distinct cell populations. Together these data demonstrate that the inflammatory environment reinforces the Treg cell-associated program.

**Inflammation-derived Treg cells have a specific effector profile**. A pairwise comparison between SF- and PB-derived Treg cells from healthy children revealed many differentially expressed genes, including core Treg cell markers including *FOXP3* and *CTLA4*, but also markers that reflect more differentiated Treg cells, like *PRDM1* (encoding Blimp-1), *ICOS, BATF*, and *BACH2* (Fig. 2a, Supplementary Data 1). Based on recent literature we analyzed the expression of markers related to eTreg cell differentiation in mice[3,13,19–34]. Hierarchical clustering analysis confirmed that Treg cells clustered separately from non-Treg cells and revealed a cluster of core Treg cell genes with increased expression in all Treg cell groups including *CTLA4*, *FOXP3*, *IL2RA* (encoding CD25), *TIGIT*, and *IKZF2* (encoding Helios) (Fig. 2b, box 1). Furthermore, clustering revealed genes that show molecular heterogeneity within the Treg cell groups. Some markers were expressed almost exclusively by SF Treg cells (including *GMZB*, *ICOS*, and *IL10*) whereas others were highly expressed by SF Treg cells but with shared, albeit lower expression by SF non-Treg cells (including *PRDM1*, *BATF*, *TNFRSF18* encoding GITR, *TNFRSF1B* and *HAVCR2* encoding TIM-3) (Fig. 2b, box 2). A third cluster concerns genes most highly expressed in non-Treg cells (Fig. 2b, box 3). Some genes in this cluster shared expression with SF Treg cells, such as *LAG3*, *CXCR3*, and *PDCD1* (encoding PD-1) whereas expression of *CD40LG*, *BACH2*, *SATB1*, *CCR7*, and *TCF7* was clearly lower in SF Treg cells (Fig. 2b, box 2 and 3). Interestingly, the more heterogeneously expressed genes listed in box 2 and 3 were previously associated with an eTreg cell profile, including *ICOS*, *IL10*, *PRDM1*, *TNRFSF18*, and *PDCD1*[3,27,30,31,33]. The increased expression of ICOS, GITR and PD-1 was verified at protein level (Fig. 2c and Supplementary Fig. 3a). Besides the differential expression of these markers, we found a significant enrichment of genes within SF Treg cells recently described in mice to be up- or downregulated in eTreg cells[35] (Fig. 2d and Supplementary Fig. 3b), confirming the eTreg cell profile. Collectively, these data demonstrate that autoimmune-inflammation-derived human Treg cells display an eTreg cell signature.

**SF Treg cells adapt to the interferon-skewed inflammatory environment**. To investigate the relationship between inflammatory environment-derived and PB-derived cells, we performed unsupervised PCA analysis on SF Treg cells and non-Treg cells,

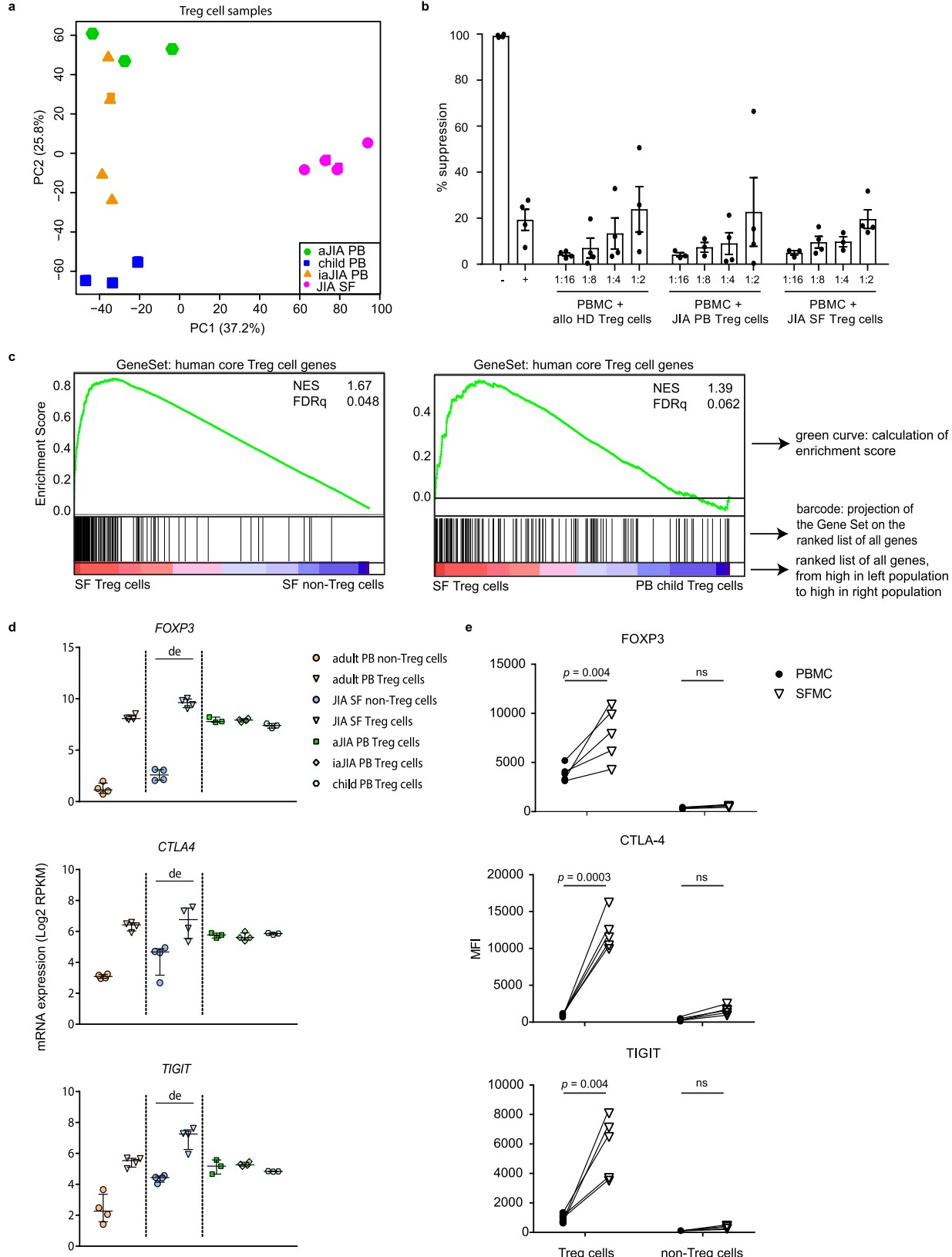

and PB Treg cells from healthy children. PB Treg cells were clearly separated from SF-derived cells, confirming that the environment plays a dominant role in determining the transcriptional landscape (Fig. 3a). K-mean clustering analysis showed that SF Treg cells and non-Treg cells share increased expression of genes related to inflammation-associated pathways,

including pathways linked to cytokine responses such as Interferon and IL-12 (Supplementary Fig. 4a). K-mean, and subsequent gene ontology analysis of SF Treg cells and all PB Treg cell groups derived from children demonstrated that pathways associated with Th1-skewing were specifically upregulated in SF Treg cells (Fig. 3b and Supplementary Fig. 4b, c). Indeed,

**Fig. 1 Distinct profile of human inflammation-derived and peripheral Treg cells. a**. Unsupervised principal component analysis (PCA) of all Treg cell groups, synovial fluid (SF)- and peripheral blood (PB)-derived, from children. **b** Quantification of suppression (percentage) of CD4$^+$ T cells by Treg cells from PB of healthy donors or juvenile idiopathic arthritis (JIA) patients and SF of JIA patients. 50,000 ctViolet labeled PBMC from an allogenic healthy donor were cultured with different ratios of Treg cells for 4 days in anti-CD3 coated plates ($n = 4$, mean ± SEM). **c** Gene set enrichment analysis (GSEA) of human core Treg cell signature genes (identified by Ferraro et al.[31]) in pairwise comparisons involving SF Treg cells and non-Treg cells, and SF Treg cells and healthy child PB Treg cells, represented by the normalized enrichment score (NES) and FDR statistical value (FDRq). **d** mRNA expression (log2 RPKM) of *FOXP3*, *CTLA4* and *TIGIT* in Treg cells derived from PB of healthy adults (adult, $n = 4$), healthy children (child, $n = 3$), JIA patients with active (aJIA, $n = 3$) or inactive (iaJIA, $n = 4$) disease, SF of JIA patients (JIA SF, $n = 4$) and non-Treg cells ($n = 3$) from PB of healthy adults and SF of JIA patients (shown are median + IQR, de = differentially expressed according to log2 fold change ≥ 0.6, adjusted (adj) *p*-value ≤ 0.05, mean of all normalized counts >10; adj *p*-values *FOXP3* = 1.1E$^{-95}$, *CTLA4* = 2.3E$^{-05}$, *TIGIT* = 1.9E$^{-17}$) **e** Median Fluorescence Intensity (MFI) of FOXP3, CTLA4 and TIGIT in CD3$^+$CD4$^+$CD25$^+$ CD127$^{low}$ Treg cells and CD3$^+$CD4$^+$CD25$^-$CD127$^+$ non-Treg cells from paired SFMC and PBMC from 5 JIA patients. Statistical comparisons were performed using two-way ANOVA with Sidak correction for multiple testing. **b**, **e** Data are representative of two independent experiments. Source data are provided as a Source Data file and deposited under GSE161426.

expression of the Th1 key transcription factor *TBX21* (T-bet), the Th1-related chemokine receptor *CXCR3*, and IL-12 receptorβ2 (*IL12RB2*) in SF Treg cells was increased on both mRNA and protein level (Fig. 3c, d and Supplementary Fig. 4d). Expression of *TBX21* and *CXCR3* was equally high in both SF Treg cells and non-Treg cells, whilst *IL12RB2* showed a significantly higher expression in SF Treg cells (adjusted *p*-value = 6.8E$^{-10}$). Accordingly, SF Treg cells in contrast to non-Treg cells showed co-expression of T-bet and FOXP3 protein excluding significant contamination of non-Treg cells potentially contributing to the high T-bet levels observed in SF Treg cells (Fig. 3e). These results indicate that SF Treg cells exhibit functional specialization to allow regulation of specific Th cell responses at particular tissue sites, as was previously established in mice[36–38]. Inline herewith, we found an enrichment in SF Treg cells of the transcriptional signature of TIGIT$^+$ Treg cells which have been identified as activated Treg cells selectively suppressing Th1 and Th17 cells in mice[19] (Supplementary Fig. 4e). The higher frequency of memory Treg cells present in SF compared to PB did not explain the differences observed (Supplementary Figs. 2d and e, 3c and 4f).

The high expression of Th1-related proteins raised the question of whether SF Treg cells may have acquired a Th1 phenotype. However, SF Treg cells failed to produce both IL-2 and IFNγ (Fig. 3f and Supplementary Fig. 4g) and responded dose-dependently to IL-2 with increasing pSTAT5 levels (Supplementary Fig. 4h), a signaling pathway pivotal for Treg cell survival and function[39]. In fact, compared to PB Treg cells, SF Treg cells appeared to be even more responsive to IL-2. Altogether, our findings demonstrate adaptation of SF Treg cells to their inflammatory environment while maintaining Treg cell key features.

**Regulation of effector Treg cells by the (super-)enhancer landscape.** To explore the mechanistic regulation of the inflammation-adapted eTreg cell profile we analyzed the enhancer landscape of SF Treg cells. Enhancers are distal regulatory elements in the DNA that allow binding of transcription factors and as such coordinate gene expression. Epigenetic regulation of enhancers is critical for context-specific gene regulation[40]. Enhancers can be defined by areas enriched for monomethylation of lysine 4 on histone H3 (H3K4me1 enrichment) and acetylation of lysine 27 on histone H3 (H3K27ac enrichment), where H3K27ac identifies active enhancers[41]. ChIP-seq performed for H3K27ac and H3K4me1 using SF Treg cells and healthy adult PB Treg cells revealed differences in enhancer profile and activity, with 3333 out of 37307 and 970 out of 10991 different peaks called between SF and PB Treg cells for H3K27ac and H3K4me1, respectively (Fig. 4a, Supplementary Fig. 5b, upper panel and Supplementary Data 2). We next assessed if the transcriptome of SF Treg cells is reflected at an epigenetic level. Genes that

demonstrated increased H3K27ac and/or H3K4me1 were increased at the mRNA level and vice versa (Fig. 4a, Supplementary Fig. 5b, middle and lowest panel and Supplementary Data 2), confirming that gene expression and chromatin-acetylation and -monomethylation are interconnected in these cells.

Super-enhancers are large clusters of enhancers that specifically regulate genes defining cell identity, both in health and disease[42,43]. Super-enhancer-associated genes were identified as the nearest TSS to the center of the enhancer and super-enhancer locus using the ROSE algorithm. Also at super-enhancer level, we found significant enrichment in the transcribed genes, again indicating that the SF Treg cell profile is mediated by epigenetic changes (Fig. 4b and Supplementary Fig. 5c, middle and lowest panel). The analysis of differential super-enhancer-associated genes revealed 337 out of 791 and 317 out of 713 different gene loci for H3K27ac and H3K4me1, respectively (Fig. 4b and Supplementary Fig. 5c, upper panel). Specifically, we identified that super-enhancers associated with genes related to canonical Th1 differentiation, such as *TBX21* and *IL12RB2*, are increased in SF Treg cells (Fig. 4c) demonstrating epigenetic regulation of the environment-associated profile. Core Treg cell genes encoding the effector molecules *ICOS*, *IL10*, *CTLA4*, *TNFRSF18*, and *PDCD1* were associated with increased super-enhancers as well as *FURIN* and *ID2* which are more putative functional Treg cell markers. eTreg cell differentiation was also reflected in the enhancer profile of SF Treg cells, with differential expression of super-enhancer-associated transcriptional regulators including *BATF*, *BACH2*, *SATB1*, *TRAF3*, and *SOCS1*. Moreover, we found super-enhancers associated with markers not previously related to (e) Treg cell differentiation and mostly specific for human Treg cells, including *VDR* (encoding the vitamin D receptor), *RXRA* (encoding Retinoic acid receptor RXR-alpha), *KAT2B* (encoding p300/CBP-associated factor (PCAF)), *TOX2* (encoding TOX High Mobility Group Box Family Member 2) and *IL12RB2*[3,4,6,44–53]. Inflammation-related homing markers (*CCR2*, *CCR5*, *CCR7*) were associated with differential H3K27ac levels in SF Treg cells as well. Finally, regulation of cell cycling and apoptosis was also mirrored in the super-enhancer landscape of SF compared to PB Treg cells by increased activity of *DPH5*, *MICAL2*, *PHLDA1*, and *CCND2*. Altogether, these super-enhancer-associated genes reflect adaptation and specialization of Treg cells within an inflammatory environment.

Motif analysis for in silico prediction of transcription factor binding sites in (super-)enhancers specifically upregulated in SF compared to PB Treg cells revealed motifs for Treg cell-specific transcription factors. These included STAT5 and Myb; the latter recently described as a core transcription factor in eTreg cell differentiation[53]. Recent papers further demonstrated BATF and RelA as crucial regulators for eTreg cells in mice[47,49]. In support

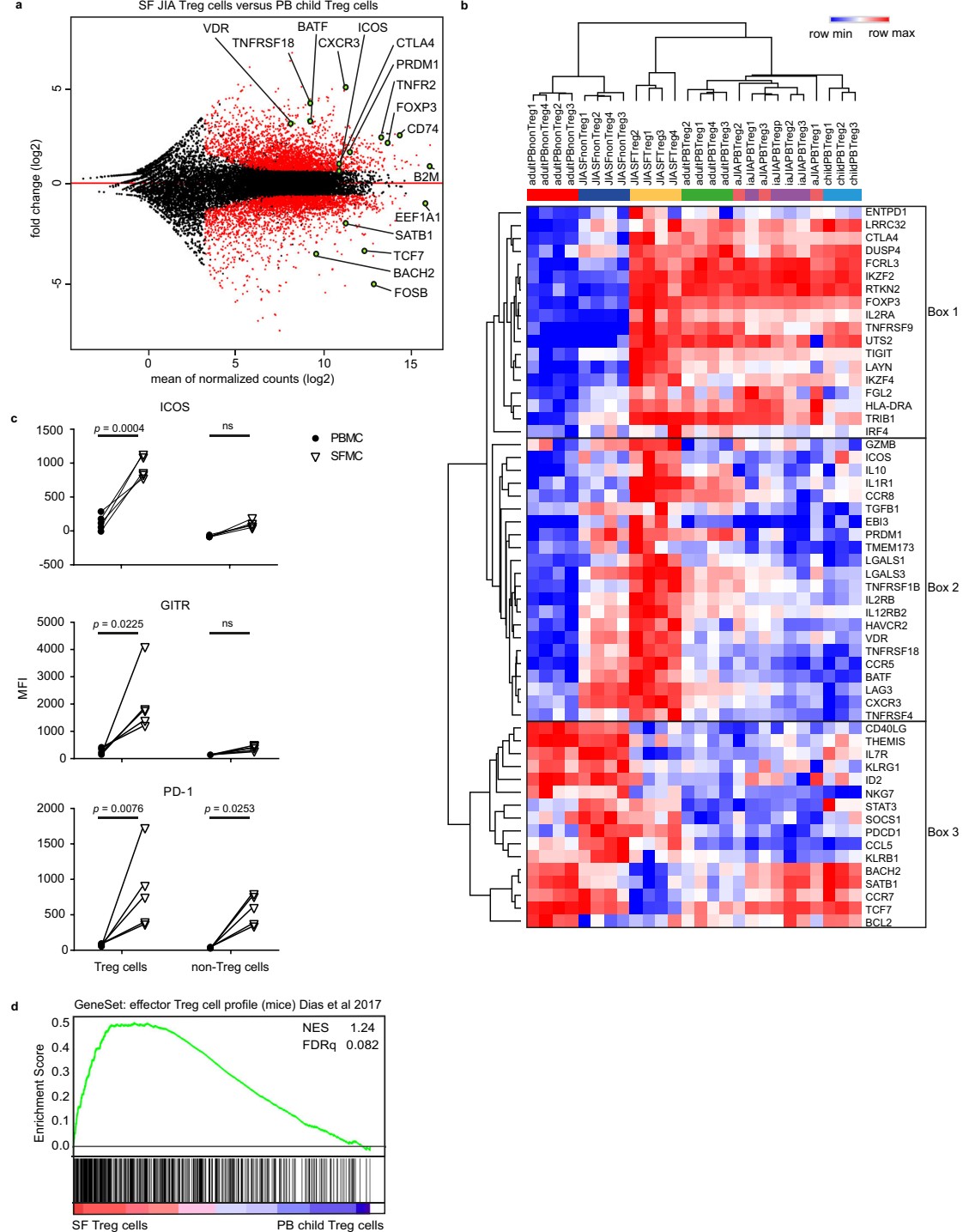

**Fig. 2 Inflammation-derived Treg cells have a specific effector profile. a** MA plot of the differentially expressed genes between synovial fluid (SF, $n = 4$) and peripheral blood (PB, $n = 3$) Treg cells of healthy children with black dots reflecting no change, and red dots transcripts with an adjusted *p*-value < 0.05 (corrected for multiple testing using the Benjamini and Hochberg method), minimal mean of all normalized counts >10 and log2 fold change > 0.6. **b** Heatmap with hierarchical clustering analysis including all groups measured with RNA-sequencing selected genes based on recent literature (relative expression of log2 RPKM). **c** Median Fluorescence Intensity (MFI) of ICOS, GITR, and PD-1 in gated $CD3^+CD4^+CD25^+CD127^{low}$ Treg cells and $CD3^+CD4^+CD25^-CD127^+$ non-Treg cells from paired SFMC and PBMC from 5 JIA patients. Statistical comparisons were performed using two-way ANOVA with Sidak correction for multiple testing. Data are representative of two or more independent experiments. **d** Gene set enrichment analysis (GSEA) of effector Treg cell genes in mice (identified by Dias et al.[53]) in pairwise comparisons involving SF Treg cells ($n = 4$) and PB Treg cells ($n = 3$) derived from healthy children, represented by the normalized enrichment score (NES) and FDR statistical value (FDRq, multiple hypothesis testing using sample permutation). Source data are provided as a Source Data file and deposited under GSE161426.

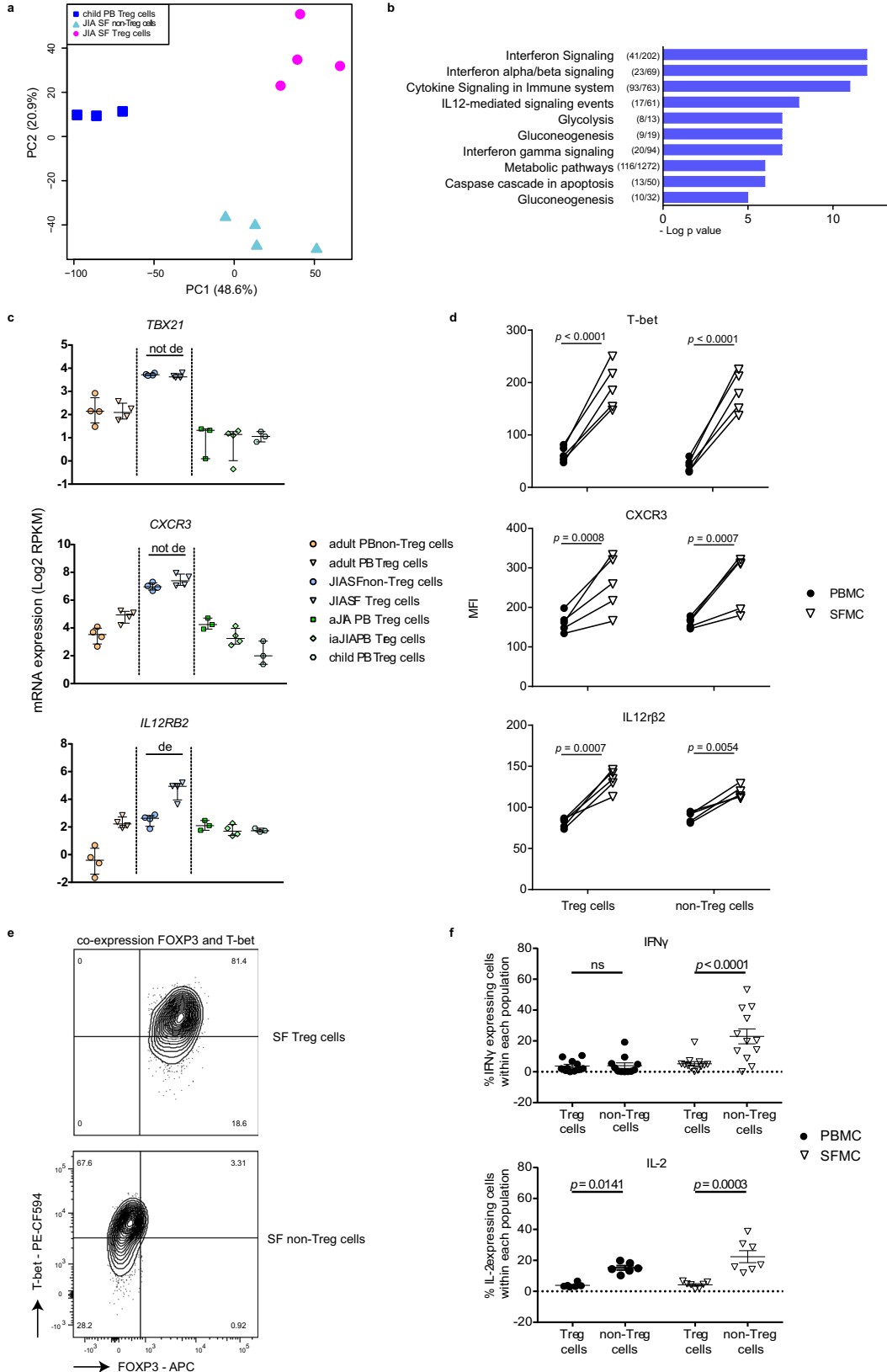

hereof, we found significant enrichment of BATF, TF65 (encoding RelA), and VDR motifs within the (super-)enhancer regions specifically upregulated in SF Treg cells (Fig. 4d and Supplementary Fig. 5d), thereby further strengthening the eTreg cell profile of SF Treg cells. The most prevalent motifs belong to the activator protein 1 (AP-1) transcription factor subfamily, part

of the basic leucine zipper (bZIP) family (Supplementary Fig. 5d) indicating this subfamily is crucial in driving the eTreg cell signature, as similarly proposed by DiSpirito et al.[54] for pan-tissue Treg cells. Moreover, within the super-enhancers of both the transcriptional regulators VDR and BATF the binding sites for each of them were found, indicating that BATF (in complex with

**Fig. 3 SF Treg cells adapt to the interferon-skewed inflammatory environment. a** Unsupervised principal component analysis (PCA) of synovial fluid (SF)-derived Treg cells and non-Treg cells (paired, $n = 4$) and peripheral blood (PB)-derived Treg cells ($n = 3$) from healthy children. **b** Gene ontology terms related to the 1791 genes specifically upregulated in SF ($n = 4$) compared to PB Treg cells derived from children (active juvenile idiopathic arthritis (JIA, ($n = 3$)), inactive JIA ($n = 4$), and healthy children ($n = 3$)), ranked by enrichment scores. The number of upregulated genes compared to the total number annotated in the gene ontology term are depicted before the terms. **c** mRNA expression (log2 RPKM) of *TBX21*, *CXCR3*, and *IL12RB2* in Treg cells derived from PB of healthy adults (adult, $n = 4$), healthy children (child, $n = 3$), JIA patients with active (aJIA, $n = 3$) or inactive (iaJIA, $n = 4$) disease, SF of JIA patients (JIA SF, $n = 4$) and non-Treg cells from PB of healthy adults ($n = 4$) and SF of JIA patients ($n = 4$) (shown are median + IQR, de = differentially expressed according to the description in Fig. 1d). Adj p-value $TBX21 = 0.91$, $CXCR3 = 0.44$, $IL12RB2 = 6.83E-10$. **d** Median Fluorescence Intensity (MFI) of T-bet, CXCR3, and IL12Rβ2 in CD3⁺CD4⁺CD25⁺CD127^low Treg cells and CD3⁺CD4⁺CD25⁻CD127⁺ non-Treg cells from paired SFMC and PBMC from 5 JIA patients. **e** Representative contourplot, for two independent experiments, of T-bet and FOXP3 in CD3⁺CD4⁺CD25⁺FOXP3⁺ Treg cells and CD3⁺CD4⁺CD25^int/⁻FOXP3⁻ non-Treg cells from SFMC. **f** Percentage of IFNγ and IL-2 positive cells measured in CD3⁺CD4⁺CD25⁺FOXP3⁺ Treg cells and CD3⁺CD4⁺CD25^int/⁻FOXP3⁻ non-Treg cells from SFMC and PMBC (IFNγ: $n = 11$ PB, $n = 12$ SF; IL-2: $n = 6$ PB, $n = 7$ SF, mean ± SEM, p-values: IFNγ PB $p > 0.9999$, IFNγ SF $p < 0.0001$, IL-2 PB $p = 0.0051$, IL-2 SF $p < 0.0001$). **d, f** Data are representative of two independent experiments. Statistical comparisons were performed using two-way ANOVA with Sidak correction for multiple testing. Source data are provided as a Source Data file and deposited under GSE161426.

other proteins) and VDR regulate their own gene expression, and also control other eTreg cell genes (Fig. 4e). Together, these observations demonstrate that the inflammation-adapted effector phenotype of SF Treg cells observed on transcriptomic level is reflected at an epigenetic level, with a super-enhancer profile and enrichment of binding sites for eTreg cell genes, showing shared features with other Treg cells in chronically inflamed tissues.

**Regulation of eTreg cell differentiation by the vitamin D3 receptor.** To extract key gene regulators, based on human transcription factors and co-factors, in the differentiation from healthy PB to SF Treg cells a data-driven network and enrichment approach was performed (RegEnrich). A network of regulators and its targets was constructed (Fig. 5a) with the top predicted regulators driving the differentiation from PB to SF Treg cells were found to be *TCF7*, *LEF1*, *JUN*, *SPEN* (negative) and *ENO1*, *THRAP3*, and *VDR* (positive) (Fig. 5a, b). *SPEN*, *THRAP3*, and *VDR* have not been previously associated with eTreg cell differentiation. *VDR* associated downstream with core Treg cell genes including *FOXP3* and *STAT3*, but also *Tbx21* (Fig. 5a, c and Supplementary Data 3). In line herewith, there was a strong positive correlation for both FOXP3 ($r = 0.772$, $p < 0.0001$) and T-bet ($r = 0.937$, $p < 0.0001$) with VDR protein expression (Fig. 5d) reinforcing the predicted regulator network.

To investigate the direct effect of vitamin D₃ on driving Treg cells towards an effector profile we incubated sorted CD3/CD28-stimulated PB Treg cells with vitamin D₃ (calcitriol/1,25-dihydroxyvitaminD3) and assessed the expression of (e)Treg cell function-related genes. Upon incubation with vitamin D₃ (10 nM; the physiological vitamin D₃ level), increased expression of *IL2RA*, *CTLA4*, *TNFRSF8* (*CD30*), and *IL10*, as well as the eTreg cell transcription factors *TBX21* and *IRF4* was observed (Fig. 5e). On protein level, CD25, CTLA4, and TNFRSF8 were also significantly increased upon vitamin D₃ incubation (Fig. 5f). In addition, in concordance with the epigenetic data, vitamin D₃ stimulation evoked a positive feedback loop on *VDR* expression (Fig. 5e). These findings support the hypothesis that vitamin D₃ signaling via VDR contributes to eTreg cell differentiation, while also being positively associated with core Treg cell marker expression.

**Similar eTreg cell profile in other inflammatory and tumor environments.** To investigate whether the program identified in SF Treg cells is more general for human Treg cells exposed to inflammation, we compared our findings with Treg cells derived from PB and inflammatory joints of rheumatoid arthritis (RA)

patients. Indeed, also in Treg cells from the inflammatory exudate of RA patients similar eTreg cell genes were upregulated (Fig. 6a).

We then compared our data to recently published human tumor-infiltrating Treg cell-specific signatures[13–15]. Strikingly, the tumor-infiltrating Treg cell signature from De Simone et al. was enriched in SF Treg cells (normalized enrichment score (NES) 1.27, FDR statistical value (FDRq) 0.071; Fig. 6b). Vice versa, the differentially expressed genes in SF Treg cells versus PB Treg cells were enriched in tumor-infiltrating compared to normal tissue Treg cells (Plitas et al., 2016: NES 1.69, FDRq 0.017, Magnuson et al., 2018: NES 1.88, FDRq 0.002; Supplementary Fig. 6a, b). Hierarchical clustering analysis of the tumor-infiltrating gene signature from De Simone et al. further revealed separate clustering of SF Treg cells (Fig. 6c), indicating that the high expression of the signature genes is specific for human Treg cells at a site that is characterized with infiltration of immune cells. We also observed a small set of genes that were not upregulated in SF Treg cells, e.g. *CX3CR1*, *IL17REL*, *IL1RL1*, and *IL1RL2*, implying environment-restricted adaptation (Fig. 6c). The three genes that were described as the most enriched and distinctive genes in tumor-infiltrating Treg cells, *LAYN*, *MAGEH1*, and *CCR8*, were selectively and highly upregulated in SF Treg cells (Fig. 6d). These data demonstrate that the Treg cell profile we observed is not restricted to the SF exudate from JIA and RA patients. In fact, it likely represents a more global profile of human Treg cells in inflammatory settings, likely fine-tuned by environment-specific adaptations.

To investigate which cues might be specific for SF we explored the differences between upregulated genes in SF and tumor-infiltrating Treg cells. Relatively few pronounced SF-specific genes were found. On gene level, cytotoxic markers including *GZMM*, *GZMA*, and *GNLY*, related to effector cell cytolysis and self-induced apoptosis of Treg cells[55] were SF-specific (Supplementary Fig. 6c). Gene ontology biological process analysis further revealed that in SF both active cell cycling and apoptotic pathways were upregulated, whereas in the tumor microenvironment cell activation and effector responses were prominent (Supplementary Fig. 6d). In addition, tumor-infiltrating Treg cells showed distinct expression of chemokines and chemokine receptors, e.g., IL-8 (CXCL8), CX3CR1, CXCR7, and CCR10 (Supplementary Fig. 6c), some of which have been proposed as a prognostic marker and/or therapeutic target in cancer[56].

From the JIA, RA, and tumor datasets we could deduce a set of genes that appear universally upregulated in eTreg cells, and also being reflected in the (super-)enhancer landscape of SF Treg cells (Fig. 6e). *ICOS*, *BATF*, *MAF*, *TNFRSF18*, and *SRGN* were revealed as core markers previously also related to mouse eTreg cells. In addition, *IL12RB2*, *VDR*, and *KAT2B*, were identified as core

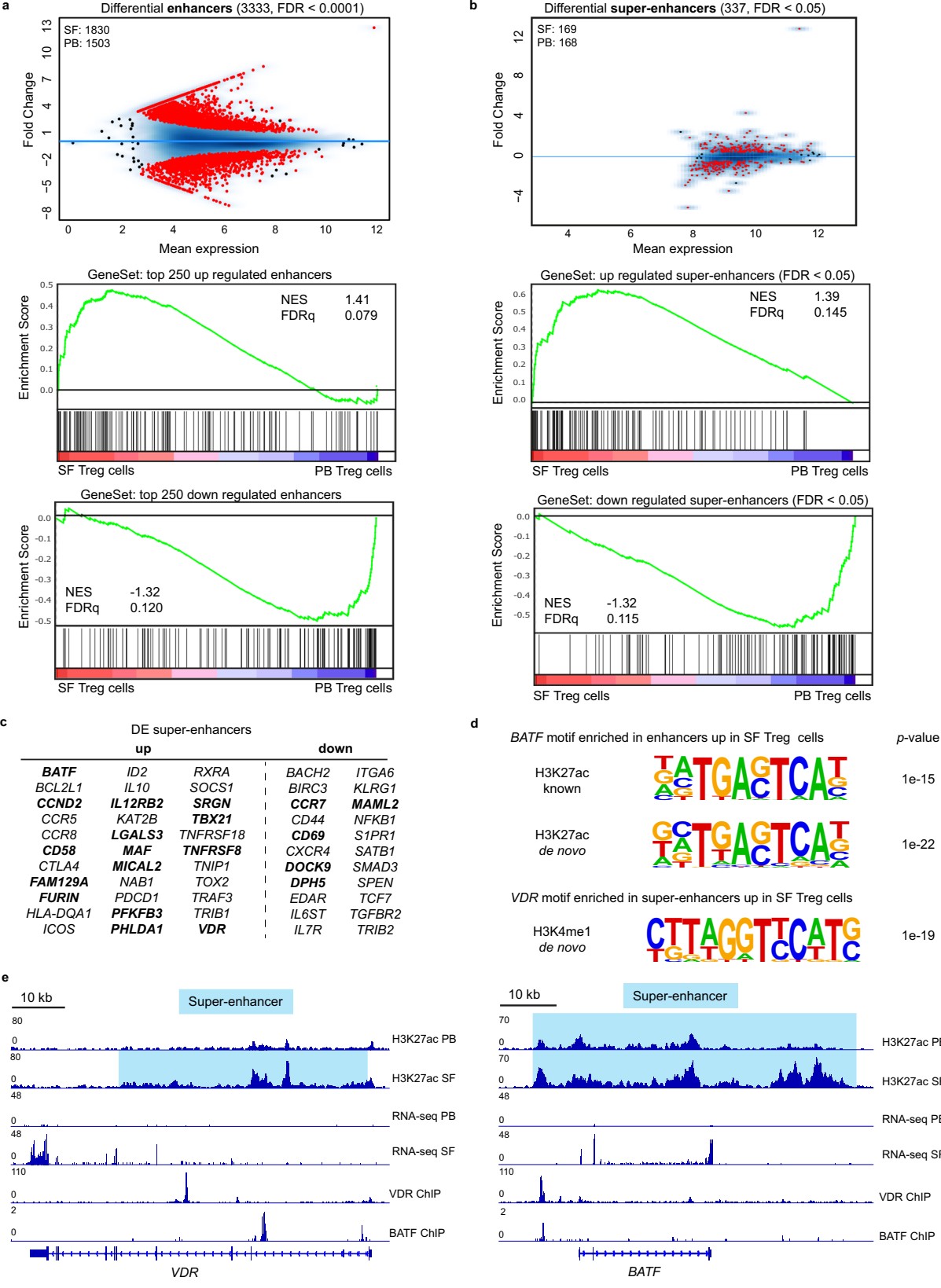

genes, as well as *PFKFB3* and *LDHA* involved in glycolysis and the apoptosis-related genes *MICAL2* and *TOX2*. The latter two have so far not been linked to Treg cells in human or mice, although both have independently been linked to human cancers[57,58]. Indeed, gene set enrichment and leading-edge analyses of activated, (shared) tissue, inflammatory and tumor-

infiltrating Treg cell gene sets amongst others revealed sharing of the core eTreg cell genes, including *BATF*, *CCND2*, *CCR8*, *TNFRSF18*, and *VDR*, (Supplementary Data 4). Even though SF Treg cells are highly activated, there was no enrichment for exhaustion nor exTreg cell-associated genes. Collectively, our results demonstrate that human Treg cells in different

**Fig. 4 Environment-specific effector Treg cell profile is regulated by the (super-)enhancer landscape. a** MA plots of differentially expressed enhancers (False Discovery Rate (FDR) < 0.0001) in synovial fluid (SF) versus peripheral blood (PB) Treg cells for H3K27ac ChIP-seq with the number of SF- and PB-specific enhancers indicated (top). Gene set enrichment analysis (GSEA) of the top 250 upregulated (middle) and downregulated (bottom) enhancers in pairwise comparisons involving transcriptome data of SF Treg cells and healthy adult PB Treg cells, represented by the normalized enrichment score (NES) and FDR statistical value (FDRq, multiple hypothesis testing using sample permutation). **b** Same as in **a** but for super-enhancers (FDR < 0.05). **c** Selection of super-enhancers up- and downregulated in SF Treg cells versus healthy adult PB Treg cells for H3K27ac and H3K4me1 ChIP-seq (FDR < 0.05; bold = up/down for both and differentially expressed (DE) in SF versus PB Treg cells on transcriptome level, see also Supplementary Data 2). **d** Motifs, known and de novo, for transcription factor binding sites predicted using HOMER, enriched in the upregulated (super-)enhancers in SF Treg cells (n = 3) compared to healthy adult PB Treg cells (n = 3) for H3K27ac and H3K4me1 ChIP-seq. p-values: cumulative binomial distribution to calculate enrichment in target versus background sequences; *BATF* H3K27ac known p = 1e−15, de novo p = 1e−22, and *VDR* H3K4me1 p = 1e−19. **e** Gene tracks for *VDR* and *BATF* (H3K27ac) displaying ChIP-seq signals, with the super-enhancer region highlighted in blue, in healthy adult PB Treg cells, SF Treg cells, VDR-specific (GSE89431), and a BATF-specific (GSE32465) ChIP-seq. RNA-seq signals for both *VDR* and *BATF* in child PB Treg cells and SF Treg cells are displayed as well. Representative samples for adult PB (n = 3), child PB (n = 3), and SF (n = 3 for ChIP-seq and n = 4 for RNA-seq) Treg cells are shown. For all panels, FDR values were calculated using the Benjamini Hochberg method unless otherwise indicated. Source data are deposited under GSE161426 and GSE156426.

inflammatory settings share an effector profile and strongly indicate that inflammation drives eTreg cell differentiation in a comparable manner in different environments.

## Discussion

The discovery of eTreg cells in non-lymphoid tissues and inflammatory sites in mice have raised questions how human eTreg cells are transcriptionally and epigenetically programmed in different settings. The present study provides evidence that human Treg cells in a local inflammatory setting undergo further differentiation into specialized eTreg cells. This is reflected by the high expression of TBX21 and CXCR3 and core Treg cell markers including FOXP3, CTLA4, and TIGIT. Essential Treg cell features are maintained, such as suppressive function, IL-2 responsiveness, and absence of cytokine production. The transcriptional changes were mirrored at an epigenetic level, both in the enhancer and super-enhancer landscape, demonstrating that the profile is highly regulated. In addition, we identify previously unappreciated and specific markers in human eTreg cells, like VDR, IL12Rβ2, and BATF; with VDR as a predicted key-regulator in human eTreg cell differentiation. In vitro, vitamin $D_3$ incubation strengthened the core (e)Treg cell signature at both the protein and mRNA level. Finally, we demonstrate that the profile is not limited to autoimmune inflammation in JIA and RA, but shares almost complete overlap with tumor-infiltrating Treg cells. The similarity between a tumor and an autoimmune setting might seem counterintuitive, since the former reflects an immune-suppressive environment whereas the latter is associated with immune activation. Both environments however share features including immune cell infiltration and inflammation[59]. This overlapping profile might therefore represent a more universal profile of human Treg cells in inflammatory environments. Because of the extensive genome-wide changes of Treg cells from an inflammatory environment insight in their functionality is important. In line with previous reports[16,17], we show that SF Treg cells are indeed suppressive and additionally highly responsive to IL-2 by phosphorylation of STAT5, excluding impaired IL-2 signaling[60,61]. This advocates against an intrinsic defect of Treg cells in inflammation. Maintenance of inflammation may be explained by the resistance of local effector cells, previously demonstrated in both JIA and RA[17,62,63].

In mice, functional specialization of Treg cells towards Th1 inflammation by upregulation of T-bet is described as essential to prevent Th1 autoimmunity[9,10]. The remarkable co-expression of T-bet and FOXP3 in SF Treg cells in our study, with T-bet levels as high as in SF non-Treg cells, implies that functional specialization is translatable to a human setting. The increased expression of migration markers such as CXCR3 and CCR5 on protein, transcriptional, and enhancer levels further strengthens this, as

high T-bet expression is crucial for migration and thus co-localization of Th1-specific Treg cells and Th1 effector T cells via upregulation of these chemokine receptors[9,10].

Contamination of Treg cells with non-Treg cells can be a concern when sorting ex vivo human cells. However, we exclude substantial contamination because (1) our data show robust transcriptomic differences between SF Treg cells and non-Treg cells, with increased expression of Treg cell hallmark genes in SF Treg cells, (2) sorted Treg cells s are > 90% FOXP3+, (3) there is clear co-expression of FOXP3 and T-bet on the single-cell level, (4) we show that SF Treg cells, while expressing high T-bet levels, lack cytokine production, potentially caused by high expression of FOXP3, SOCS1, EGR2, and EGR3 (Supplementary Data 1)[64,65]. Nevertheless, as recently shown by Miragaia et al.[4] there may be heterogeneity within the eTreg cell population.

Our study investigated both the transcriptional and epigenetic regulation of eTreg cells in human inflammatory settings allowing us to discern human-specific eTreg cell regulation as well as commonalities between mice and humans. The findings on epigenetic level should still be demonstrated experimentally since although regions marked by H3K27ac/H3K4me1 are associated with active enhancers this is not prima facie evidence of enhancer function[41,66]. Furthermore, we have identified (super-)enhancer associated genes as the nearest TSS to the center of the enhancer and super-enhancer locus. It is therefore possible that these (super-)enhancers can regulate other genes than predicted by our analyses[41]. This requires further studies into the spatial chromatin organization in Treg cells, especially under inflammatory conditions. The epigenetic landscape of human Treg cells is little explored, with only two papers studying H3K27acetylation and H3K4monomethylation of circulating human Treg cells[67,68]. Arvey et al. conclude that the majority of Treg cell lineage-specific elements are not conserved between mice and human further emphasizing the lack of knowledge of the human Treg cell epigenetic landscape. Gao et al. focused on SNPs within enhancers of PB Treg cells from Type 1 diabetes patients, and in line with our data showed, that cell cycle and apoptosis regulation is highly reflected in the enhancer landscape of these patients. These data further underscore the need for additional studies, especially in non-lymphoid tissues during non-homeostatic conditions. We found similarities between mice and humans in eTreg cell differentiation regarding downregulation of *SATB1*, *BACH2*, *TCF7*, and *LEF1*, all associated with a super-enhancer in PB but not SF Treg cells. Although all four genes are crucial for Treg cell development, recent reports show that downregulation of these regulators is necessary for further differentiation of Treg cells and preventing conversion into Th cells[44,52,69–73]. BACH2 can transcriptionally repress *PRDM1* (encoding Blimp-1) in T cells[72], and in line herewith we observed increased acetylation and gene

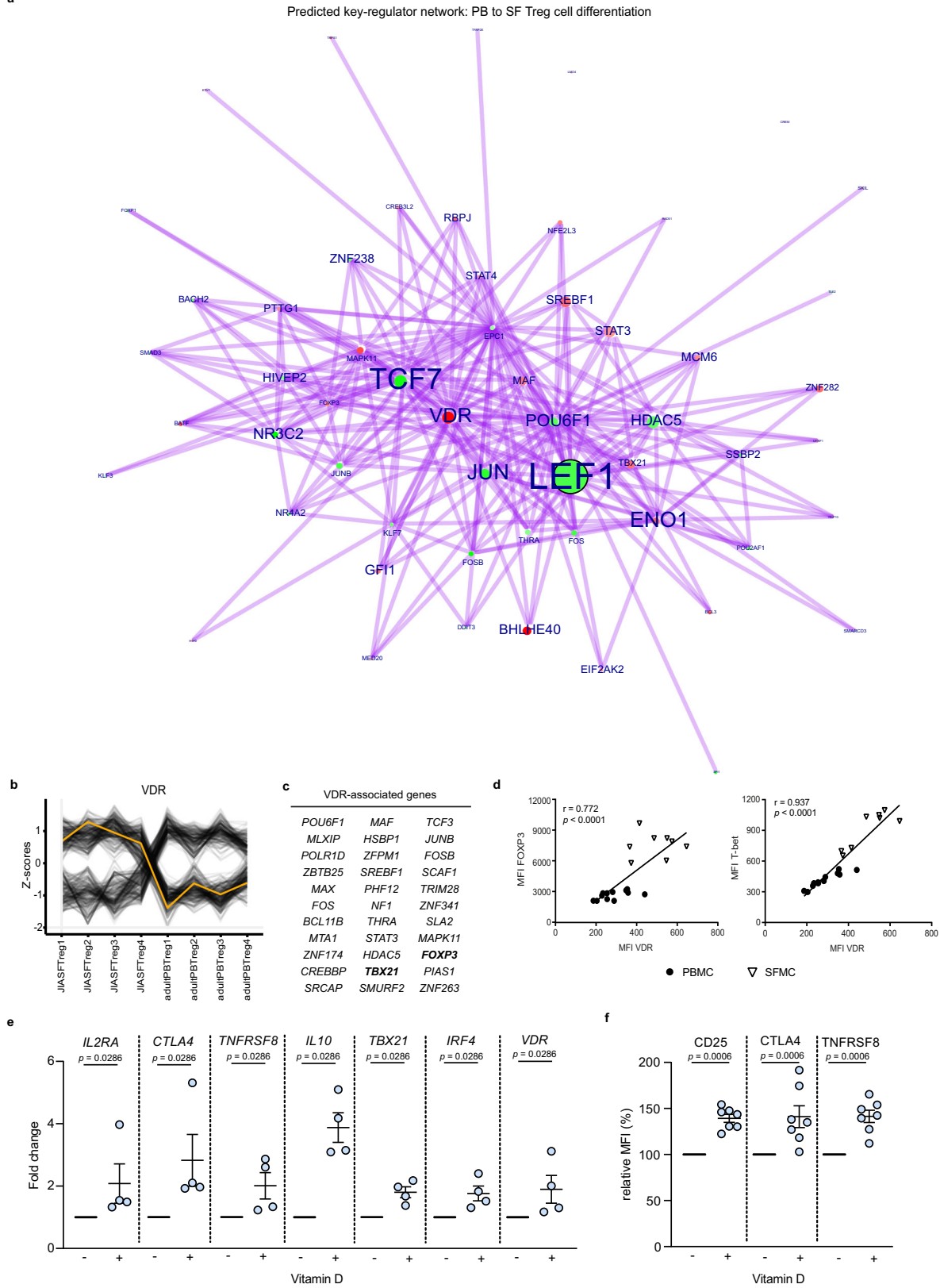

**b** VDR

**c** VDR-associated genes

| POU6F1 | MAF | TCF3 |
|---|---|---|
| MLXIP | HSBP1 | JUNB |
| POLR1D | ZFPM1 | FOSB |
| ZBTB25 | SREBF1 | SCAF1 |
| MAX | PHF12 | TRIM28 |
| FOS | NF1 | ZNF341 |
| BCL11B | THRA | SLA2 |
| MTA1 | STAT3 | MAPK11 |
| ZNF174 | HDAC5 | **FOXP3** |
| CREBBP | **TBX21** | PIAS1 |
| SRCAP | SMURF2 | ZNF263 |

**d** 

r = 0.772
p < 0.0001

r = 0.937
p < 0.0001

● PBMC   ▽ SFMC

**e** 

Fold change

IL2RA  p = 0.0286
CTLA4  p = 0.0286
TNFRSF8  p = 0.0286
IL10  p = 0.0286
TBX21  p = 0.0286
IRF4  p = 0.0286
VDR  p = 0.0286

Vitamin D

**f** 

relative MFI (%)

CD25  p = 0.0006
CTLA4  p = 0.0006
TNFRSF8  p = 0.0006

Vitamin D

expression of *PRDM1* in SF Treg cells. This was further supported by the decrease in *TCF7* and *LEF1*, both negatively correlating with eTreg cell marker expression including *PRDM1*[73]. Moreover, we found BATF as a prominent marker in SF Treg cells, with increased expression on both transcriptional and epigenetic level, and a binding site in upregulated super-enhancer associated

genes. In addition, high BATF expression was also observed in RA SF Treg cells and tumor-infiltrating Treg cells, suggesting BATF is a key transcriptional regulator in both human and mice eTreg cells. We also found remarkable differences in eTreg cell signature markers compared to what is known from mice. Most pronounced is upregulation of IL12RB2 at both the

**Fig. 5 The vitamin D receptor is a predicted regulator of effector Treg cell differentiation. a** Network inference of key-regulators driving peripheral blood (PB) to synovial fluid (SF) Treg cell differentiation on RNA level, based on unsupervised weighted correlation network analysis followed by Fisher's exact test of the transcription factors and co-factors (red = upregulation, green = downregulation). The purple lines depict connections between regulators and their targets. Circle size indicates $-\log 10(p)$ for each comparison, with the $p$(-value) derived from differential expression analysis (log fold change > 1), and the text size represents the RegEnrich score; for both, larger indicates higher scores (see also Supplementary Data 3). **b** The RNA expression profile of *VDR* from JIA SF Treg cell samples 1–4 to adult PB Treg cell samples 1–4 indicated by the yellow line, with its associated genes in gray; values are normalized to the Z-score. **c** All genes associated with VDR defined by the key-regulator network inference. **d** Pearson's correlation plot of VDR and FOXP3 (left) and T-bet (right) Median Fluorescence Intensity (MFI's) in PB (circles) and SF (triangles) with the line fitted by linear regression ($n = 12$ PB HC and PB JIA (PBMC), $n = 8$ SF JIA (SFMC)). **e** Relative **e**xpression fold change ($2^{\Delta\Delta CT}$) of *IL2RA*, *CTLA4*, *TNFRSF8*, *IL10*, *TBX21*, *IRF4*, and *VDR* (lower left panel) on Treg cells upon incubation of 50,000 activated CD3$^+$CD4$^+$CD25$^+$CD127$^{low}$ sorted Treg cells from HC PB with 10 nM vitamin D compared to without vitamin D for 2 days ($n = 4$, mean ± SEM). **f** Relative MFI changes of CD25 (IL2RA), CTLA4, and TNFRSF8 following experiment as per **e** with ($n = 7$, mean ± SEM). **d**–**f** Data are representative of two independent experiments. Source data are provided as a Source Data file.

transcriptional and epigenetic level, and a binding site in upregulated super-enhancer associated genes, whereas in mice eTreg cells this marker is downregulated. Instead, impaired IL-12Rβ2 expression has been reported as a key checkpoint preventing Treg cells from fully differentiating towards Th1 cells in mice[38]. However, our data are in line with recent reports showing selective expression of *IL12RB2* in human PB Treg cells[30,67], suggesting this marker has different functions in human and mice.

Another unexpected finding is the upregulation of VDR in both JIA and RA SF Treg cells, on both transcriptional and epigenetic level, as well as in tumor-infiltrating Treg cells. Although not highlighted, high VDR levels were present in breast tumor-infiltrating Treg cells[14] and uterine eTreg cells[74]. Besides the well-known tolerogenic effects of vitamin D₃[75], VDR is not well-studied in the Treg cell context, especially in humans. A recent study did show that memory CCR6$^+$ Th cells gained a suppressive phenotype, including CTLA4 expression, and functional suppressive capacity similar to Treg cells upon incubation with vitamin D₃[76]. Here we show VDR is a predicted key-regulator in SF eTreg cells, positively correlated with both FOXP3 and T-bet expression.

Another difference is the absence of *KLRG1* upregulation in human eTreg cells. In mice, KLRG1 is a key marker to identify eTreg cells, although not crucial for eTreg cell function[77]. KLRG1 is not upregulated on SF Treg cells and associated with a super-enhancer only in PB Treg cells. In addition, in our data gene expression of KLRG1 is restricted to non-Treg cells (both SF and PB). Confusing the findings are discrepancies possibly caused by the tissue measured or comparison made. Miragaia et al.[4] showed upregulation of *PIM1* in non-lymphoid Treg cells compared to PB Treg cells in mice, but *PIM1* was not expressed in human tissue Treg cells. In our data however, *PIM1* is highly increased in SF compared to PB Treg cells of healthy adults and children (Supplementary Data 1). In summary, we show that human eTreg cells show similarities in eTreg cell differentiation markers identified in mice, but also found human-specific eTreg cell markers including IL-12Rβ2 and VDR, and absence of KLRG1.

Previous research has shown that ex vivo human Treg cells are highly glycolytic[78], in line with their high proliferative capacity[79]. In mice increased glycolysis is required for eTreg cell differentiation and migration to inflammatory sites[80,81]. In support hereof, glycolysis-associated genes including *PFKFB3*, *LDHA*, and *PKM2* were increased in SF Treg cells on both RNA and (super-)enhancer levels. Finally, *ENO1*, encoding the glycolytic enzyme Enolase 1, is a predicted key-regulator of eTreg cell differentiation.

Lately, the application of Treg cell-based therapies for autoimmune diseases and transplantation settings is gaining renewed interest. Promising data from animal models and clinical trials, both with cell therapy involving the adoptive transfer and with low dose IL-2 administration, pave the way for Treg cell therapies

to reach the clinic[82]. Our data show that circulating human Treg cells are markedly different from their counterparts derived from sites characterized by immune activation. This is reflected in the expression of effector markers and transcriptional regulators, but also in the expression of specific chemokine receptors. Regarding Treg cell-based therapies appreciating that specific environments may require adapted Treg cells, both for migration and function, is important. Moreover, in this study, we profiled human eTreg cells abundantly present at sites of inflammation. This information can form a basis for follow-up studies that may eventually allow the characterization of small amounts of circulating eTreg cells, for example to monitor patients undergoing treatment.

In conclusion, our study uncovered the transcriptional and epigenetic program that defines human inflammatory Treg cells. SF Treg cells display an environment-adapted as well as an eTreg cell phenotype established at the epigenetic level. Moreover, we describe striking similarities of the eTreg cell program with tumor-infiltrating Treg cells and SF Treg cells from RA patients. This revealed a set of genes shared with human eTreg cells from affected sites in JIA, RA, and cancer including BATF, VDR, MICAL2, TOX2, KAT2B, PFKFB3, and IL12Rβ2. Finally, we show that THRAP3, ENO1, and VDR are predicted key-regulators driving human SF eTreg cell differentiation at the transcriptomic level.

## Methods

**Collection of SF and PB Samples.** Patients with JIA ($n = 41$) were enrolled by the Pediatric Rheumatology Department at University Medical Center of Utrecht (The Netherlands). Of the JIA patients $n = 8$ were diagnosed with extended oligo JIA and $n = 33$ with oligo JIA, according to the revised criteria for JIA[83], with an average age of 11.3 years (range 3.2–19 years) and disease duration at the time of inclusion of 4.9 years (range 0.1–15 years). Active disease was defined by physician global assessment of ≥1 active joint (swelling, limitation of movement), and inactive disease was defined as the absence hereof. Additional parameters including the Juvenile Arthritis Disease Activity Score (JADAS), C-reactive protein (CRP), and erythrocyte sedimentation rate (ESR) were also taken into account. Patients with RA ($n = 7$) were enrolled by the rheumatology outpatient clinic at Guy's and St. Thomas' Hospital NHS Trust (United Kingdom). The average age of RA patients was 61 years (range 30–75 years). PB and SF was obtained when patients visit the outpatient clinic via vein puncture or intravenous drip, and by therapeutic joint aspiration of the affected joints, respectively. The study was conducted in accordance with the Institutional Review Board of the University Medical Center Utrecht (approval no. 11-499/C; JIA) and the Bromley Research Ethics Committee (approval no. 06/Q0705/20; RA). PB from healthy adult volunteers (HC, $n = 20$, average age 41.7 years with range 27–62 years) was obtained from the Mini Donor Service at University Medical Center Utrecht. PB from $n = 8$ healthy children (average age 11.4 years with range 7.3–15.6 years; approval no. 05-149/K) was obtained from a cohort of control subjects for a case-control clinical study. Informed consent was obtained from all the participants and/or from their parents/guardians/legally authorized representatives.

**Isolation of SFMC and PBMC.** SF of JIA patients was incubated with hyaluronidase (Sigma-Aldrich) for 30 min at 37 °C to break down hyaluronic acid. SF mononuclear cells (SFMCs) and peripheral blood mononuclear cells (PBMCs) were isolated using Ficoll Isopaque density gradient centrifugation (GE Healthcare Bio-

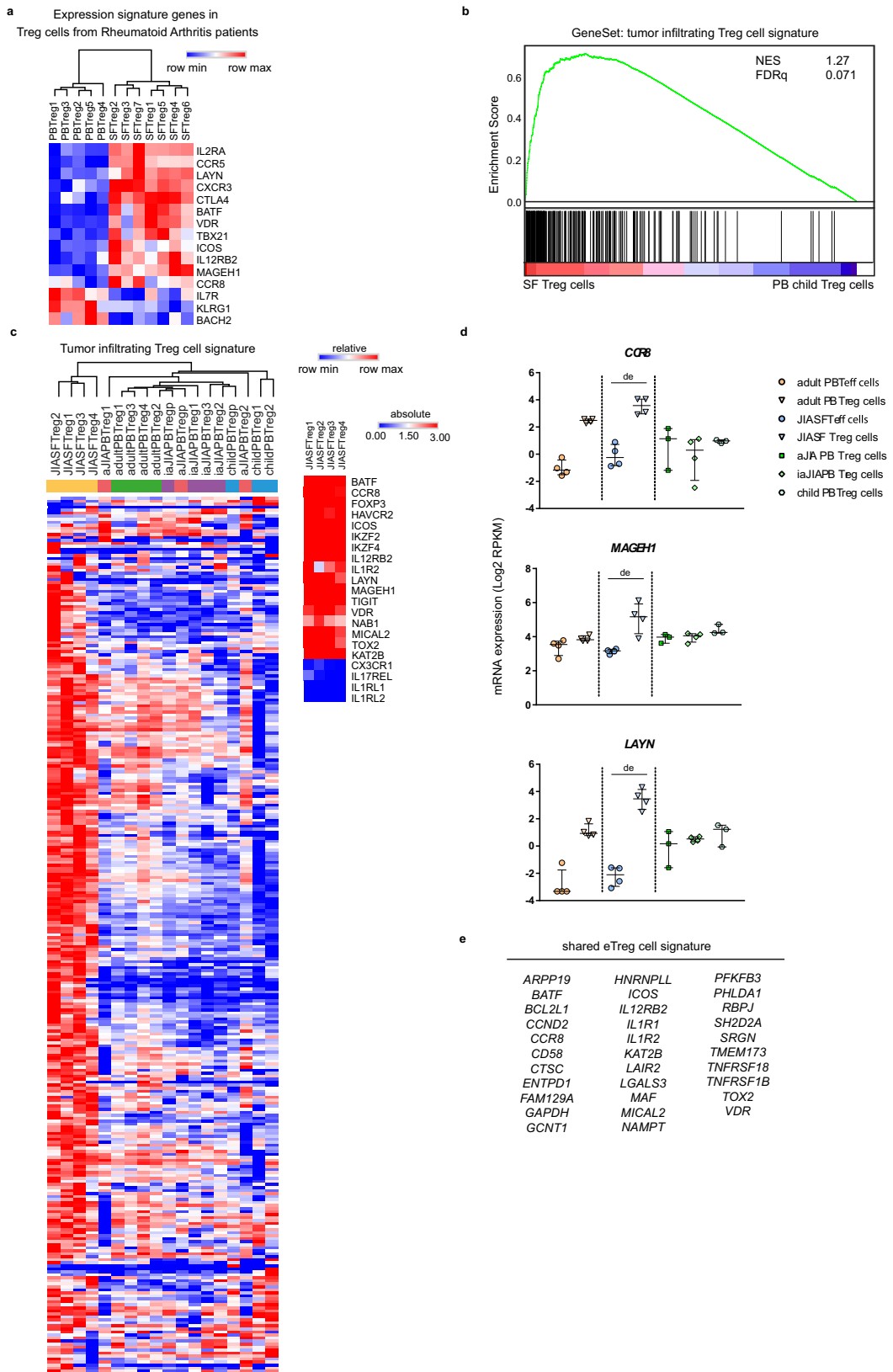

Sciences, AB) and were used after freezing in fetal calf serum (FCS) (Invitrogen) containing 10% DMSO (Sigma-Aldrich).

**Suppression assay**. CD3+CD4+CD25+CD127low cells (Treg cells) were isolated from frozen PBMC (see Supplementary Fig. 1a for gating strategy), using the FACS Aria III (BD). Antibodies used for sorting are: anti-human CD3-BV510 (clone

OKT3, 1:400), CD25-PE/Cy7 (clone M-A251, 1:25; BD), CD127-AF647 (clone HCD127, 1:50; Biolegend), CD4-FITC (clone RPA-T4, 1:200; eBioscience). To check for FOXP3 expression of the sorted populations cells were fixed and permeabilized by using eBioscience Fixation and Permeabilization buffers (Invitrogen) and stained with anti-human FOXP3-eF450 (clone PCH101, 1:50; eBioscience) (see Supplementary Fig. 1b). Read out of proliferation is performed with the following antibodies: CD3-PerCP/Cy5.5 (clone UCHT1, 1:100; Biolegend), CD4-FITC (clone

**Fig. 6 The effector Treg cell and human tumor Treg cell signature overlap. a** Heatmap with unsupervised hierarchical clustering analysis on peripheral blood (PB, $n = 5$) and (partially paired) synovial fluid (SF, $n = 7$) Treg cells from rheumatoid arthritis patients measured with a gene array, on selected signature genes identified from juvenile idiopathic arthritis (JIA) SF Treg cells and tumor-infiltrating Treg cells. **b** Gene set enrichment analysis (GSEA) of tumor-infiltrating Treg cell signature genes (identified by De Simone et al.[13]) in pairwise comparisons involving SF Treg cells ($n = 4$) and healthy child PB Treg cells ($n = 3$), represented by the normalized enrichment score (NES) and FDR statistical value (FDRq, multiple hypothesis testing using sample permutation). **c** Heatmap with hierarchical clustering analysis including all groups measured with RNA-sequencing on the identified tumor-infiltrating Treg cell signature genes (ref. [13]), with relative expression of log2 RPKM. Small heatmap on the right shows log2 RPKM values of a selection of genes in SF Treg cells. **d** mRNA expression (log2 RPKM) of *CCR8*, *MAGEH1* and *LAYN* in Treg cells from PB of healthy adults (adult, $n = 4$), healthy children (child, $n = 3$), JIA patients with active (aJIA, $n = 3$) or inactive (iaJIA, $n = 4$) disease, SF of JIA patients (JIA SF, $n = 4$) and non-Treg cells from PB of healthy adults ($n = 4$) and SF of JIA patients ($n = 4$) as determined by RNA-sequencing analysis (shown are median + IQR, de = differentially expressed according to the description in Fig. 1d; adjusted *p*-values (Benjamini and Hochberg method) *CCR8* = 2.2E−13, *MAGEH1* = 1.2E−07, *LAYN* = 1.2E−33). **e** Identification of a human effector Treg cell profile based on the overlapping genes upregulated in JIA SF Treg cells, RA SF Treg cells, and tumor-infiltrating Treg cells (identified in several studies, see refs. [13–15,89,90]), and reflected on the (super-)enhancer landscape of SF Treg cells. Source data are provided as a Source Data file and deposited under GSE161426.

RPA-T4, 1:200; eBioscience), CD8-APC (clone SK1, 1:100; BD). Total PBMC were labeled with 2 μM ctViolet (Thermo Fisher Scientific) and cultured alone or with different ratios of sorted Treg cells (1:16, 1:8, 1:4, 1:2). Cells were cultured in RPMI1640 media containing 10% human AB serum with the addition of L-Glutamine and Penicillin/Streptomycin. PBMC were stimulated by 0,1 μg/ml coated anti-CD3 (eBioscience) and incubated for four days in a 96 well round bottom plate (Nunc) at 37 °C. After 4 days cells were stained with CD3, CD4, and CD8 for read out of proliferation by flow cytometry performed on FACS Canto II (BD Biosciences) and data were analyzed using FlowJo Software v10 (Tree Star Inc.).

**Cell phenotyping**. PBMC and SFMC were thawed and stained by surface and intranuclear/cellular staining with the following antibodies: CD3 BV510 (clone OKT3, 1:400), CD3 AF700 (clone UCHT1, 1:50), CD127 PerCP-Cy5.5 (clone HCD127, 1:25), CD127 PE-Cy7 (clone HCD127, 1:50), CXCR3 FITC (clone G025H7, 1:40), IL2 PB (clone MQ1-17H12, 1:100; Biolegend), CD4 APC-eFluor780 (clone RPA-T4, 1:50), FOXP3 APC (clone PCH101, 1:25), TIGIT PerCP-eFluor710 (clone MBSA43, 1:50), FOXP3 eF450 (clone PCH101, 1:50), ICOS APC (clone ISA3, 1:20; eBioscience), CD127 BV421 (clone HIL-7R-M21, 1:40), CD25 PE/Cy7 (clone MA251, 1:25), CTLA4 PE (clone BNI3, 1:12.5), PD-1 BV711 (clone EH12.1, 1:100), IL12Rβ2 PE (clone 2B6/12β2, 1:5), T-bet PE-CF594 (clone 04-46, 1:25), IFNγ PerCP-Cy5.5 (clone 4S.B3, 1:40), FOXP3 PE-CF594 (clone 259D/C7, 1:50; BD), GITR FITC (clone #110416, 1:8.3; R&D). For measurement of cytokines thawed cells were stimulated with 20 ng/mL PMA (MP Biomedicals) and 1 μg/mL ionomycin (Calbiochem) in RPMI supplemented with 10% AB. After 30 min incubation at 37 °C Golgi stop (BD Biosciences) was added followed by 3.5 h of incubation at 37 °C. For intranuclear/cellular staining the Intracellular Fixation & Permeabilization Buffer Set (eBioscience) was used. See Supplementary Fig. 1a and c for gating strategy of FOXP3+ Treg cells and non-Treg cells, and Supplementary Figs. 2–4 for MFI plots of the markers of interest. Data acquisition was performed on a BD LSRFortessa (BD Biosciences) and analysis as above.

**STAT5 Phosflow**. PBMC and SFMC were thawed and resuspended in PBS ($0.5-1.0 \times 10^6$ living cells/tube). Surface staining of CD3-BV510 (clone OKT3, 1:400, Biolegend) and CD4-FITC (clone RPA-T4, 1:200, eBioscience) was performed for 25 min at 4 °C. Cells were then stimulated with 0, 1, 10 or 100 IU/ml human (h)IL-2 (Proleukin; Novartis) for 30 min at 37 °C, fixated and permeabilized by using buffers from the Transcription Factor Phospho Buffer Set (BD Biosciences). Intranuclear staining of FOXP3-eF450 (clone PCH101, 1:50), T-bet-eF660 (clone eBio4B10, 1:50; eBioscience), CD25-PE/CY7 (clone M-A251, 1:25) and pSTAT5-PE (clone pY695, 1:25; BD) was performed for 50 min at 4 °C. See Supplementary Fig. 1a and c for gating strategy of FOXP3+ Treg cells and non-Treg cells, and Supplementary Fig. 4h for the gating of pSTAT5. Data acquisition and analysis as above.

**RNA-sequencing**. CD3+CD4+CD25+CD127low and CD3+CD4+CD25−CD127+ cells were sorted by flow cytometry from HC PBMC and JIA patient SFMC and PBMC (see Supplementary Fig. 1a for gating strategy). Total RNA was extracted using the AllPrep DNA/RNA/miRNA Universal Kit (Qiagen) as specified by the manufacturer's instructions and stored at −80 °C. Sequencing libraries were prepared using the Rapid Directional RNA-Seq Kit (NEXTflex). Libraries were sequenced using the Nextseq500 platform (Illumina), producing single end reads of 75 bp (Utrecht Sequencing Facility). Sequencing reads were mapped against the reference human genome (hg19, NCBI37) using BWA (v0.7.5a, mem −t 7 −c 100 −M -R) (see Supplementary Data 5 for quality control information). Differential gene expression was performed using DEseq2 (v1.2). For K-means clustering and PCA analysis, genes with fold change between samples on 10th and 90th quantile at least 1 log2 RPKM and expression at least 2 log2 RPKM in the sample with the

maximal expression were used. K-means clustering was done on gene expression medians per group, with an empirically chosen k of 14. The scripts used for analysis are available at[84]. Gene Ontology pathway analyses were performed using ToppFun (https://toppgene.cchmc.org/enrichment.jsp) with input genes belonging to the defined k-mean clusters, with an FDR-corrected *p*-value < 0.05 defining significance. Gene set enrichment analysis (GSEA, v3.0, and v4.0.3)[85], with as input the log2 RPKM data, was used to assess whether specific signatures were significantly enriched in one of the subsets. One thousand random permutations of the phenotypic subgroups were used to establish a null distribution of enrichment score against which a NES and FDR-corrected q values were calculated. Gene-sets were either obtained by analyzing raw data using GEO2R (NCBI tool) or downloaded from published papers, or self-made based on the H3K27ac/H3K4me1 data. In particular, the following published gene data sets were used: human core Treg cell signature: Ferraro et al.[31]; effector Treg cell signature in mice: Dias et al.[53]; tumor-infiltrating Treg cell signature: De Simone et al.[13]. Plitas et al.[14]. Magnuson et al.[15]; effector Treg cell genes in mice: Levine et al.[35]; TIGIT+ Treg cell signature in mice: Joller et al.[19]. Identification of key-regulators was performed using RegEnrich[86] v1.0.0 based on the differential gene expression data followed by unsupervised WGCNA for the network inference, mean RPKM counts >0 were included, and Fisher's exact test was used for enrichment analysis. Heatmaps and subsequent hierarchical clustering analyses using One minus Pearson correlation were performed using Morpheus software (v0.1.1.1, https://software.broadinstitute.org/morpheus/).

**H3K27ac and H3K4me1 ChIP-sequencing**. PBMC from HC and SFMC from JIA patients were thawed and 0.5–1 million CD3+CD4+CD25+CD127low cells were sorted by flow cytometry (see Supplementary Fig. 1a for gating strategy). For each sample, cells were crosslinked with 2% formaldehyde and crosslinking was stopped by adding 0.2 M glycine. Nuclei were isolated in 50 mM Tris (pH 7.5), 150 mM NaCl, 5 mM EDTA, 0.5% NP-40, and 1% Triton X-100 and lysed in 20 mM Tris (pH 7.5), 150 mM NaCl, 2 mM EDTA, 1% NP-40, 0.3% SDS. Lysates were sheared using Covaris microTUBE (duty cycle 20%, intensity 3, 200 cycles per burst, 60-s cycle time, eight cycles) and diluted in 20 mM Tris (pH 8.0), 150 mM NaCl, 2 mM EDTA, 1% X-100. Sheared DNA was incubated overnight with anti-histone H3 acetyl K27 antibody (ab4729; Abcam) or anti-histone H3 (mono methyl K4) antibody (ab8895; Abcam) pre-coupled to protein A/G magnetic beads. Cells were washed and crosslinking was reversed by adding 1% SDS, 100 mM NaHCO3, 200 mM NaCl, and 300 μg/ml proteinase K. DNA was purified using ChIP DNA Clean & Concentrator kit (Zymo Research), end-repair, a-tailing, and ligation of sequence adaptors were done using Truseq nano DNA sample preparation kit (Illumina). Samples were PCR amplified, checked for the proper size range and for the absence of adaptor dimers on a 2% agarose gel and barcoded libraries were sequenced 75 bp single-end on Illumina NextSeq500 sequencer (Utrecht DNA sequencing facility). Sample demultiplexing and read quality assessment was performed using Base-Space (Illumina) software. Reads with quality score of Q > 30 were used for downstream analysis. Reads were mapped to the reference genome (hg19/hg38) with Bowtie 2.1.0 using default settings for H3K27ac ChIP-seq and BWA for H3K4me1 ChIP-seq (see Supplementary Data 5 for quality control information). SAM files were converted to BAM files using samtools version 0.1.19. Peaks were subsequently called using MACS-2.1.0. Enriched regions were identified compared to the input control using MACS2 callpeak --nomodel --exttsize 300 --gsize=hs -p 1e-9. The mapped reads were extended by 300 bp and converted to TDF files with igvtools-2.3.36 and were visualized with IGV-2.7.2[87] for H3K27ac and H3K4me1 ChIP-seq. Differential binding analysis was performed using the R package Diff-Bind v1.8.5. In DiffBind read normalization was performed using the TMM technique using reads mapped to peaks which were background subtracted using the input control. Enhancer gene associations were determined as the nearest TSS to the center of the enhancer and super-enhancer locus. Super-enhancers were identified by employing the ROSE algorithm[88] using a stitching distance of the MACS2 called peaks of 12.5 kb, peaks were excluded that were fully contained in

the region spanning 1000 bp upstream and downstream of an annotated TSS (-t 1000). The H3K27ac/H3K4me1 signal was corrected for background using the input control and subsequently ranked by increasing signal (Fig. S5A). Super-enhancer gene associations were determined as the nearest TSS to the center of the enhancer and super-enhancer locus using the ROSE algorithm. BEDtools v2.17.0 was used for general manipulation of peak bed-files. Motif enrichment analysis was performed using the HOMER software v4.11 (findMotifsGenome.pl; hg19/hg38; -size 200). ChIP-seq data for activated Treg cells, VDR, and BATF were retrieved from GEO:GSE43119 (GSM1056948 and GSM1056952), GSE89431 (GSM2371449) and GSE32465 (GSM803538), respectively.

**VDR and 1,25 vitamin D3 incubation assay**. Thawed HC PBMC and JIA patient SFMC were stained by surface and intranuclear staining as described above with the following antibodies: anti-human fixable viability dye eF506 (1:1000), FOXP3-PerCP/Cy5.5 (clone PCH101, 1:50; eBioscience), CD3-AF700 (clone UCHT1, 1:50), CD4-BV785 (clone OKT4, 1:100; Biolegend), CD25-BV711 (clone 2A3, 1:50), CD127-BV605 (clone A019D5, 1:50; Sony Biotechnology), and VDR-PE (clone D-6, 1:100; Santa Cruz Biotechnology). In addition, CD4+ T cells were isolated from fresh PBMC using the CD4+ T cell isolation kit and LS column (Miltenyi Biotec). Sorted Treg cells (CD3+CD4+CD25+CD127low, 50.000) were plated in a round bottom 96-wells plate in the presence of 10% human AB serum, anti-CD3/CD28 (Dynabeads Human T-activator CD3/CD28, Thermo Fisher Scientific) at a 1:5 ratio (1 bead to 5 cell ratio), 100 IU/ml (h)IL-2 (Proleukin; Novartis) for 2 days. In addition, 0 (control) or 10 nM 1α,25-Dihydroxyvitamin D3 (Sigma-Aldrich) was added. Expanded Treg cells were stained by surface and intranuclear staining as described above with the following antibodies: anti-human fixable viability dye eF506 (1:1000), FOXP3-eF450 (clone PCH101, 1:50; eBioscience), CD3-AF700 (clone UCHT1, 1:50), CD4-APC/Cy7 (clone RPA-T4, 1:100; Biolegend), CD25-BV711 (clone 2A3, 1:50), and CD30-FITC (clone Ber-H8, 1:12.5), and CTLA4-APC (clone REA945, 1:50; Miltenyi Biotec), CD30-FITC (clone BNI3, 1:12.5; BD). See Supplementary Figs. 1–2 for the gating strategy. Data acquisition and analysis as above. Furthermore, total RNA from expanded Treg cells was extracted using the PicoPure RNA Isolation Kit (Thermo Fisher Scientific) and RNA was reverse transcribed using the iScript cDNA Synthesis Kit (Bio-Rad). Quantitative RT-PCR was performed using SYBR Select Master Mix (Life Technologies) according to the manufacturer's instructions and measured on a QuantStudio 12k Flex real-time PCR (Thermo Fisher Scientific). Fold change was calculated using the delta-delta Ct method ($2^{\Delta\Delta Ct}$) normalized to the housekeeping gene GUSB in Microsoft Excel 2016. The primers used are listed in Supplementary Table 1.

**Microarray RA Treg cells**. CD14+ monocytes (purity > 98%) were depleted through positive selection using CD14 MicroBeads (Miltenyi Biotec). CD4+ T cells were isolated from the CD14− cell fraction by negative selection (Miltenyi Biotec) and stained with CD4-PerCP/Cy5.5 (clone SK3, 1:200), CD45RA-APC/Cy7 (clone HI100, 1:50), CD45RO-PB (clone UCHL1, 1:50), CD127-FITC (clone A019D5, 1:50; Biolegend), and CD25-PE (clone REA945, 1:50; Miltenyi Biotec). CD4+CD45RA−CD45RO+CD25+CD127low cells (memory Treg cells, see Supplementary Fig. 1a for gating strategy)), were sorted using a BD FACSAria II. Sorted Treg cell samples were lysed in 1000 μl of TRIzol (Invitrogen). Chloroform (200 μl) was added, and the samples were then whirl mixed and incubated for 2–3 min at room temperature. Following centrifugation (10.000 × g for 15 min at 4 °C), the water phase was further purified using the ReliaPrepTM RNA Miniprep system (Promega) per manufacturer's instructions. RNA integrity was confirmed on an Agilent Technologies 2100 bioanalyzer. One hundred nanograms of total RNA was used to prepare the targets (Affymetrix) in accordance with the manufacturer's instructions. Hybridization cocktails were hybridized onto a Human Gene 2.0 ST Array. Chips were scanned and gene expression data were normalized using the RMA algorithm. Gene expression analysis was performed using Qlucore Omics Explorer software, version 3.0.

**Statistics and reproducibility**. For ChIP-seq and RNA-seq analysis, p-values were adjusted with the Benjamini-Hochberg procedure. Protein and cytokine expression was analyzed with Pearson's correlation, two-tailed Mann–Whitney $U$ test, Two-way ANOVA with Sidak correction for multiple testing or a mixed-effects model with Dunnett post hoc on paired data with missing values using GraphPad Prism. All experimental flow cytometry and sorting data concern two or more independent experiments with similar results. RNA- and ChIP-sequencing runs were performed in one batch.

**Reporting summary**. Further information on research design is available in the Nature Research Reporting Summary linked to this article.

## Data availability
RNA-seq and ChIP-seq data generated for this study have been deposited in the Gene Expression Omnibus (GEO) database under the accession codes: GSE161426 and GSE156418. ChIP-seq data for activated Treg cells, VDR and BATF was retrieved from GEO:GSE43119 (GSM1056948 and GSM1056952), GSE89431 (GSM2371449) and GSE32465] (GSM803538), respectively. Raw data for figures are provided in the Source Data file. All other data are available in the manuscript and its Supplementary information.

## Code availability
The code used for RNA-seq analysis are available at https://github.com/mmokry/Mijnheer_Nat_Commun_2021[84].

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

## Acknowledgements

We would like to thank J. van Velzen and P. Andriessen van der Burght for technical assistance, and Anoushka Samat for help with qPCR primer design. F. van Wijk is supported by a VIDI grant from ZonMw (91714332).

## Author contributions

Conceptualization: G.M., L.L., Fv.W., and J.v.L. Performed experiments: G.M., L.L. M.v.d. W., R.S., M.K., S. Vervoort, and A. Petrelli, Data analysis: G.M., L.L., M.M., M.v.dW., J. G.C.P., A. Pandit, W.T., M.W., and J.v.L. Patient selection and clinical interpretation: G. M., L.L., S. Vastert, and S.d.R. Supplied RA dataset: V.F., C.R., and L.S.T. Writing: G.M., L.L., M.v.d.W., J.v.L., and F.v.W. Supervision: J.v.L and F.v.W. All authors reviewed and edited the manuscript.

## Competing interests

The authors declare no competing interests.
