## [Peer Review File · Nature Communications]

REVIEWER COMMENTS

Reviewer #1 (Remarks to the Author):

The authors of this study have presented novel data regarding characterization of T regulatory cells (Tregs) in synovial fluids of patients with inflammatory arthritis. The findings are of interest because Tregs are major regulators of immune responses, and an understanding of these cells and their related pathways has implications for management of autoimmune and inflammatory conditions. Strengths of the study include the multifaceted approach, including gene expression, flow cytometric analyses and functional assays. Statistical analyses are appropriate. Tregs from SF show clear phenotypic differences when compared to those from PB. Furthermore, samples are obtained from patients with arthritis, including paired peripheral blood (PB)/synovial fluid (SF) samples, another strength. Some weaknesses of the study are also noted. One is that functional properties in terms of suppression and cytokine production are similar in PB and SF, which brings into question the significance of the phenotypic differences in these two subsets. Plasticity of Tregs in conditions of inflammation are well-known; a question is whether the alterations that are described lessen the regulatory effects, thus being more permissive of the ongoing inflammation. The authors propose that responses of effector cells are diminished, but whether that is a key mechanistic contribution is not clear. Specific points follow:

1. RA synovial fluid consists primarily of polymorphonuclear leukocytes, with T cells being less prevalent, so the absolute number of T regs might be small and thus make a relatively minor contribution to immune processes within the joint. Supplementary Fig 1 does show that Tregs are a greater proportion of the CD3/CD4 population than the corresponding PB, but whether this makes a significant quantitative contribution to SF Tregs in the synovial space is somewhat unclear. Also, the suppressive function comparison for SF and PB Tregs using equal numbers for the in vitro cultures (Figure 1) may not be representative of the in vivo conditions.
2. A related issue is that the functional characteristics of SF and PB Tregs appear to be similar in terms of suppressive capabilities (Figure 1b) and lack of production of the cytokines IL-2 and IFN gamma (Figure 3F), which brings into question how the phenotypic differences impact Treg functions in the synovial space.
3. Plasticity of Tregs in the inflammatory environment is well-described, but whether the changes that are described are more consistent with enhanced or diminished regulatory function is not clearly addressed. Maintenance of joint inflammation may be due to resistance of local effector cells, as proposed (pp 14-15), but this appears somewhat speculative.
4. The JIA patients are described as being clinically active or inactive (results, line 100, in figure legends and figures), but no description of how this activity status was determined is provided.
5. Methods for measurement of cytokines in Figure 3f are not provided; were the cells stimulated in culture? Also, in the legend for this figure, line 850, the words "positive cells" may be needed following "IL-2."
6. Patients with JIA and RA have significant differences, with the latter usually being associated with the presence of autoantibodies, while the former may be mediated through different immune and inflammatory pathways and without autoantibodies. This does not necessarily lessen the usefulness of comparing these diseases, but consideration might also be given to other conditions that are inflammatory but not autoimmune, such as crystalline or infectious arthritides as useful comparisons. The finding that RA SF Tregs have characteristics similar to those in JIA SF extends and somewhat generalizes the results, but whether this is sufficient to consider the pattern "universal" for Tregs under conditions of inflammation may somewhat overstate the case (page 11; line 245).
7. Similarities with tumor-infiltrating Tregs are of interest in the context of longstanding data regarding the tissue invasive properties of synovial pannus cells in adult RA, and study of Tregs in synovial tissues from such patients would be of interest in this regard.
8. The HC adult population had an average age that was 20 years less than that of the RA patients; it would be useful show that the data in these two groups did not exhibit age-related differences, to strengthen the validity of the comparison.
9. The program used to analyze proliferation data by flow cytometry is not described.

Reviewer #2 (Remarks to the Author):

The aim of this study was to identify a core transcriptional signature of human effector regulatory T cells (eTregs) that are derived from an autoimmune/inflammatory environment.

The authors used juvenile idiopathic arthritis (JIA) as well as rheumatoid arthritis (RA) as model diseases and explored the transcriptional and epigenetic profiles of Tregs derived from synovial fluid (SF) of these patients. The transcriptional profile of SF Tregs was assessed by RNA sequencing (and RNA microarrays) and analyzed using PCA, hierarchical clustering analysis as well as gene set enrichment analysis. Data from SF Tregs were compared to that of SF and peripheral blood (PB) Tregs and non-Tregs from patients and healthy controls as well as to published data sets derived from tumor-infiltrating Tregs. The authors described a specific effector profile in SF Tregs that is characterized by exclusive expression of GMZB, ICOS and IL10 and increased expression of PRDM1, BATF, GTR (amongst other markers). This core profile was present in JIA and RA SF Tregs as well as tumor derived Tregs. Using gene ontology analysis the authors demonstrated increased expression of interferon and IL-12 related genes in SF Tregs (e.g. TBX21/T-bet and IL12RB2). The authors additionally analyzed the enhancer landscape (H3K4me1 and H3K27ac enrichment) using ChIP-sequencing and could demonstrate close correlation between the transcriptional and the epigenetic profile within SF Tregs. Additionally, the authors tried to predict regulators that might drive the differentiation of SF eTregs and suggested the vitamin-D receptor (VDR) to be a key regulator of eTreg differentiation.

This is an interesting study and albeit being mainly descriptive adds to our understanding of eTreg differentiation in an inflamed/autoimmune environment. The strength of the study is the comparative analysis of Treg transcriptional profiles derived from PB and inflamed tissue (SF) from defined inflammatory/autoimmune diseases. The use of two data sets from different inflammatory/autoimmune diseases (JIA and RA) is appreciated. The manuscript is well written, the study is well designed and the figures are excellent.

I have the following questions/comments:

-The manuscript could benefit from functional validation, especially of the "predicted key-regulator in eTreg differentiation" VDR. Did the authors try to assess the in vitro impact of vitamin D on the induction of the eTreg expression signature?

- References are missing for mentioned literature (p.6, line 126 „Based on recent literature ...“ and p.6 line 139 „... were previously associated with an eTreg profile ...“)

- Figure 3:d The differences in expression of the analyzed proteins (T-bet, CXCR3, IL12RB2) and particularly the expressed cytokines IFN- γ and IL-2 at least in non-Tregs might be biased by a higher frequency of memory T cells within the SF compartment that are per se enriched in e.g. cytokine expressing/polarized cells

- Figure 4: The authors should comment why PB Treg from healthy adults were used for epigenetic analysis whereas PB Treg from children (controls and JIA) were used for transcriptional analysis

- p. 9, line 206: "we found super enhancers associated with markers not previously related to (e)Treg differentiation and mostly specific for human Treg ..." -> quote references

- Materials and Methods: the description of flow cytometric analysis regarding analysis of protein expression in SF Treg is almost completely missing, e.g. antibodies used for analysis of CTLA-4, TIGIT, PD1, ICOS ... and stimulation procedures and analysis of cytokine expression (IL-2 and IFN- γ)

- p.24, line 539: "Vitamin D receptor and 1,25 vitamin D3 incubation assay" -> no incubation assay is described in this paragraph. Has this been performed?

Reviewer #3 (Remarks to the Author):

I want to thank the authors and editors for the opportunity to review this manuscript by Mijnheer and colleagues. In this paper, the authors show that that FOXP3+ T cells (Tregs cells) isolated from synovial fluids of children with juvenile idiopathic arthritis (and adult RA) show a distinctive epigenetic and expression profile that strongly resemble so-called effector Tregs (eTreg) previously identified in mice. A refreshing aspect to this paper is the fact that the authors recognize that the differences between the Tregs seen in the peripheral blood of children with JIA and those identified in their synovial fluid are likely a reflection of normal homeostatic mechanisms. The pediatric and adult rheumatology literature is, unfortunately, awash in papers where this idea is not considered, and where the authors have concluded that differences between peripheral blood and synovial fluid cells is *prima facie* evidence of "dysregulation" of the synovial fluid cells. I have a few comments and suggestions that I hope the authors will find helpful.

1. Throughout the paper, the authors assume that H3K4me1/H3K27ac-marked regions are functioning enhancers. While these chromatin marks provide strong evidence that the region of interest has enhancer function, such function must be demonstrated experimentally (eg., Kessler et al, PLoS One 2020). It might be awkward to call the H3K4me1/H3K27ac-marked regions "putative enhancers" or "likely enhancers" everywhere they are referred to in the text, but some acknowledgement early in the discussion that H3K4me1/H3K27ac marks are not *prima facie* evidence of enhancer function would enhance the scientific rigor of the paper.
2. I'm not sure at all what the authors mean by "super-enhancer associated genes." Are they referring to the genes most proximal to the enhancers? This term may be applicable to intronic enhancers, which often regulate the gene in which they are located, and they show such enhancers in Figure 4e. However, in the broader context, enhancers often regulate genes many kB away, and not always the most proximal genes. However, Gasperini et al (Cell 2019) have shown that >70% of enhancers regulate genes within the same chromatin loop of topologically associated domain (TAD). Because we still have an incomplete catalogue of what genes are regulated by what enhancers, it's hard to know what the authors' term "enhancer-associated genes" means. This reviewer would suggest that lines 192-214 be re-written.
3. The points being made in Figure 4e would have been more clear if the authors had also provided the RNAseq track with the ChIPseq tracks.
4. Line 229 - "...these observations demonstrate that the inflammation-adapted effector phenotype...is regulated at the epigenetic level." I'm wondering, "As opposed to what?" What was the alternative if not epigenetic re-organization? Indeed, we're learning that for terminally-differentiated cells, re-organization of enhancers (and therefore epigenetic profiles) is the mechanism through which fine-tuned adaptation to local environments occur (e.g., Gosselin et al, Cell 2014). The key finding in this paper isn't that the transcriptional and functional differences between PB and SF Tregs are associated with re-organization of H3K4me1/H3K27ac marks. That was entirely predictable. What's interesting is that these cells share common features with other Tregs in other chronically inflamed tissues.
5. In that line, it would be intriguing to see whether the CD4/FoxP3+ cells from other chronically inflamed tissues share similar features to the cells described in this paper. The most accessible for individuals in a large children's hospital might be the CD4+ T cells found in the lungs of patients with cystic fibrosis.

Some minor comments

1. This reviewer was surprised to see the word "men" to describe humans (lines 12 and 342). In my own institution, this would be considered exclusive and a faux pas. I am aware that the senior author, Dr van Wijk, is a talented and respected scientist who is also a woman.
2. Some of the expansive language was a little off-putting for this reviewer, e.g., "eminently" (p. line 69) "highly relevant" (line 71). My own view is that, if you need an adverb to modify an adjective, you might want to select another adjective. I think this rule works in any Indo-European language.

James N Jarvis

Reviewer #4 (Remarks to the Author):

The manuscript by Mijnheer et al. examined transcriptional and epigenetic profiles of the human effector Tregs, and showed the unique transcriptional patterns and similarities to other types of Tregs such as tumor-infiltrating Tregs. Based on the analysis the authors claimed that VDR functions as a key-regulator in effector Treg differentiation; however, simple comparisons only enumerate the possibilities. Conclusions should be drawn through experimental confirmations.

In several cases, the authors claimed the similarities between SF Tregs and selected characteristics via GSEA. GSEA can examine the similarities with cells of your own choosing only. More objective views or a comparison with multiple cells would be required. For example, it has been reported that a variety of Tregs, such as effector Tregs, naïve Tregs, exhausted Tregs, exTregs, tissue-resident Tregs, activated Tregs, tumor-infiltrating Tregs were present in our bodies. Therefore, the authors should examine the similarities of SF Tregs to these Tregs, and then claim that one specific population shows the highest similarities to SF Tregs.

The authors claimed that VDR functions as a key-regulator in effector Treg differentiation. Does VDR repression (for example, via Crisper-Cas9, siRNA, KD) interfere the differentiation of effector Tregs? Conversely, does Vitamin D enhance the differentiation? Such experimental confirmation would be essential.

The authors also claimed that autoimmune inflammation-derived Tregs display a unique transcriptional profile. However, FOXP3, CTLA-4, TIGIT, GITR, etc. would be upregulated in Tregs by TCR stimulation (CD3/CD28 antibody stimulation in vitro), and also tissue-resident Tregs express BATF. TCR stimulation-dependent transcriptional changes and changes in tissue-resident cells would be included in the comparison.

Minor comments:

In Fig. 1b, control samples are required in the Treg suppression assay.

This reviewer could not understand the graphical details and an importance of Fig. 5a. Improvement would be required.

We would like to thank the reviewers for their constructive comments on our manuscript. We performed additional experiments and have adjusted our manuscript based on the feedbacks suggested. Specifically we have now:

- Performed additional experiments and included these new data demonstrating that Vitamin D promotes the (e)Treg core program (Fig. 5e and f).*
- Performed new analyses with CD45RO Treg (Supplementary Figure 2d, 3c and 4e)*
- Added RNA-seq tracks (Fig. 4e).*
- Performed a broad comparison of all 369 immunological signatures concern both mouse and human Treg (GSEA as well as leading edge analysis Supplementary Table 4)*

Please find below a point-by-point reply to all questions and comments. We have adapted the manuscript according to the reviewers comments. Substantial adaptations have been marked in red.

REVIEWER COMMENTS

Reviewer #1 (Remarks to the Author):

The authors of this study have presented novel data regarding characterization of T regulatory cells (Tregs) in synovial fluids of patients with inflammatory arthritis. The findings are of interest because Tregs are major regulators of immune responses, and an understanding of these cells and their related pathways has implications for management of autoimmune and inflammatory conditions. Strengths of the study include the multifaceted approach, including gene expression, flow cytometric analyses and functional assays. Statistical analyses are appropriate. Tregs from SF show clear phenotypic differences when compared to those from PB. Furthermore, samples are obtained from patients with arthritis, including paired

peripheral blood (PB)/synovial fluid (SF) samples, another strength. Some weaknesses of the study are also noted. One is that functional properties in terms of suppression and cytokine production are similar in PB and SF, which brings into question the significance of the phenotypic differences in these two subsets. Plasticity of Tregs in conditions of inflammation are well-known; a question is whether the alterations that are described lessen the regulatory effects, thus being more permissive of the ongoing inflammation. The authors propose that responses of effector cells are diminished, but whether that is a key mechanistic contribution is not clear. Specific points follow:

We thank the reviewer for the time and advice, and are happy to respond to these questions.

1. RA synovial fluid consists primarily of polymorphonuclear leukocytes, with T cells being less prevalent, so the absolute number of Tregs might be small and thus make a relatively minor contribution to immune processes within the joint. Supplementary Fig 1 does show that Tregs are a greater proportion of the CD3/CD4 population than the corresponding PB, but whether this makes a significant quantitative contribution to SF Tregs in the synovial space is somewhat unclear. Also, the suppressive function comparison for SF and PB Tregs using equal numbers for the in vitro cultures (Figure 1) may not be representative of the in vivo conditions.

We agree with the reviewer that other immune cells also contribute to the inflammation and that this may be one of the reasons why, despite their relative enrichment, Treg numbers are still insufficient to suppress a full-blown inflammatory response (Buckner et al. 2010 Nat Rev Immunol; Morgan et al 2003 & 2005 Arthritis Rheum; Ehrenstein et al. 2004 J Exp Med; Nguyen et al. Arthritis Rheum 2007). It has long been speculated that Treg from an inflammatory environment may lose their suppressive capacity. We and others have however shown that SF-derived Treg are functional, but that effector T cells become unresponsive to suppression due to hyperactivation of the protein kinase B (PKB) pathway (van Amelsfort et al. 2004 and Wehrens et al. 2011).

The aim of Figure 1 was to confirm that the suppressive capacity of these SF Tregs was not impaired compared to blood Treg, hence similar Treg numbers were used. These experiments demonstrated that the Treg present in synovial fluid are true Treg by expression of FOXP3 as well displaying suppressive capacity.

2. A related issue is that the functional characteristics of SF and PB Tregs appear to be similar in terms of suppressive capabilities (Figure 1b) and lack of production of the cytokines IL-2 and IFN gamma (Figure 3F), which brings into question how the phenotypic differences impact Treg functions in the synovial space.

We and others demonstrated that effector T cells become resistant to suppression by Treg in SF, with the local milieu and specifically IL-6/TNF α enhancing activation of protein kinase B (PKB)/c-akt in effector cells (van Amelsfort et al. 2004 and Wehrens et al. 2011). While it is hard to mimic the in vivo situation with in vitro suppression assays what we clearly demonstrate is:

- 1) Enrichment and expansion of stable Treg in SF, with an enhanced core signature at the gene, protein and epigenetic level.*
- 2) Despite their effector differentiation, SF Treg have not lost their suppressive capabilities*
- 3) SF Treg have acquired an inflammation-induced phenotype that allows active recruitment (including upregulation of CXCR3 as shown in Fig. 3d and CCR5)*

3. Plasticity of Tregs in the inflammatory environment is well-described, but whether the changes that are described are more consistent with enhanced or diminished regulatory function is not clearly addressed. Maintenance of joint inflammation may be due to resistance of local effector cells, as proposed (pp 14-15), but this appears somewhat speculative.

We observe higher expression of core Treg gene and protein expression including FOXP3, CTLA4, ICOS, TIGIT, all of which have been demonstrated to contribute to Treg suppressive function, indicating that SF Treg have a stable phenotype.

Resistance of effector cells to suppression by Treg has been shown in many autoimmune diseases including Type I diabetes (Schneider et al. 2008 J Immunol; Lawson et al. 2008 Clin Exp Immunol), Psoriasis (Goodman et al. 2009 J Immunol), Crohn's disease (Fantini et al. 2009 Gastroenterology) and SLE (Venigalla et al. 2008 Arthritis Rheum, Vargas-Rojas et al. 2008 J Lupus). Also in JIA, previous publications have shown that resistance of local effector cells to Treg-mediated suppression due to TNF α /IL-6 induced hyperactivation of the PKB-pathway contributes to the maintenance of joint inflammation (Wehrens et al. 2011; van Amelsfort et al. 2004 and Petrelli et al., Arthritis Rheumatol 2016).

4. The JIA patients are described as being clinically active or inactive (results, line 100, in figure legends and figures), but no description of how this activity status was determined is provided.

Clinical active disease was defined by physician global assessment of ≥ 1 active joint (swelling, limitation of movement), and clinically inactive disease was defined as the absence hereof. Additional parameters including the Juvenile Arthritis Disease Activity Score (JADAS), C-reactive protein (CRP) and erythrocyte sedimentation rate (ESR) were also taken into account. We now state this in the method section (line 442-446).

5. Methods for measurement of cytokines in Figure 3f are not provided; were the cells stimulated in culture? Also, in the legend for this figure, line 850, the words "positive cells" may be needed following "IL-2."

We thank the reviewer for the attentiveness and have edited the line to include 'positive cells', and included the stimulation procedure in the method section (line 491-495).

6. Patients with JIA and RA have significant differences, with the latter usually being associated with the presence of autoantibodies, while the former may be mediated through different immune and inflammatory pathways and without autoantibodies. This does not necessarily lessen the usefulness of comparing these diseases, but consideration might also be given to other conditions that are inflammatory but not autoimmune, such as crystalline or infectious arthritides as useful comparisons. The finding that RA SF Tregs have characteristics similar to those in JIA SF extends and somewhat generalizes the results, but whether this is sufficient to consider the pattern "universal" for Tregs under conditions of inflammation may somewhat overstate the case (page 11; line 245).

Indeed we have assessed Treg isolated from the inflamed joints of both JIA and RA patients. Our observations correlate with a study comparing osteoarthritis and rheumatoid arthritis which showed that Treg in SF in both settings were of a memory phenotype, highly activated and expressed similar protein levels of CTLA4, PD-1 and GITR (Moradi et al., Arthritis

Research & Therapy 2014). We agree that our wording “universal” is a bit strong and have adapted the manuscript accordingly.

7. Similarities with tumor-infiltrating Tregs are of interest in the context of longstanding data regarding the tissue invasive properties of synovial pannus cells in adult RA, and study of Tregs in synovial tissues from such patients would be of interest in this regard.

We completely agree with the reviewer that a study of Treg in synovial tissues would be of interest. In the past two years multiple studies have been published that focused on synovium tissue in patients with inflamed joints, primarily RA patients (Zhang et al., Nat Immunol 2019; Stephenson et al., Nat Com 2018; Rao et al., Nature 2017). Although none of the studies have focused on Treg in synovium, single cell studies data point to the presence of adapted Treg. In the study of Zhang et al a Treg cluster is identified of which the top 20 genes defining that Treg cluster are also separating SF Treg from SF non-Treg, PB adult Treg and/or PB child Treg in our study (for example: FOXP3, TIGIT, CTLA4, IKZF2, IKZF4, DUSP4, F5, and RGS1). It thus seems there are similarities in adaptation of invasive Treg in the synovium of RA patients compared to the synovial fluid, as has been shown for effector T cells.

8. The HC adult population had an average age that was 20 years less than that of the RA patients; it would be useful show that the data in these two groups did not exhibit age-related differences, to strengthen the validity of the comparison.

The heatmap displayed in Fig. 6a show PB Treg derived from 5 of the 7 RA patients of which we also have SF Treg, so it regards (partially) paired data and a difference in age is not of concern. This is stated in the figure legend of Figure 6a, result section (line 258-261) and the method section (line 448-450).

For the comparison of SF Treg derived from JIA patients we had a healthy adult population that was older, but also included age-matched healthy control PB Treg to rule out age effects.

9. The program used to analyze proliferation data by flow cytometry is not described.

We have included additional information regarding all our flow cytometry data, including the proliferation data. Our analyses were performed using Flowjo v10 (Tree Star Inc.; line 480 and ‘data analysis as above’ with the subsequent method sections employing Flowjo).

Additionally, we have now included representative Flowjo histogram plots to visualize gating of proliferation/deduction of suppression (Supplementary Fig. 2a).

Reviewer #2 (Remarks to the Author):

The aim of this study was to identify a core transcriptional signature of human effector regulatory T cells (eTregs) that are derived from an autoimmune/inflammatory environment.

The authors used juvenile idiopathic arthritis (JIA) as well as rheumatoid arthritis (RA) as model diseases and explored the transcriptional and epigenetic profiles of Tregs derived from synovial fluid (SF) of these patients. The transcriptional profile of SF Tregs was assessed by RNA sequencing (and RNA microarrays) and analyzed using PCA, hierarchical clustering analysis as well as gene set enrichment analysis. Data from SF Tregs were compared to that of SF and peripheral blood (PB) Tregs and non-Tregs from patients and healthy controls as well as to published data sets derived from tumor-infiltrating Tregs. The authors described a

specific effector profile in SF Tregs that is characterized by exclusive expression of GMZB, ICOS and IL10 and increased expression of PRDM1, BATF, GITR (amongst other markers). This core profile was present in JIA and RA SF Tregs as well as tumor derived Tregs. Using gene ontology analysis the authors demonstrated increased expression of interferon and IL-12 related genes in SF Tregs (e.g. TBX21/T-bet and IL12RB2). The authors additionally analyzed the enhancer landscape (H3K4me1 and H3K27ac enrichment) using ChIP-sequencing and could demonstrate close correlation between the transcriptional and the epigenetic profile within SF Tregs. Additionally, the authors tried to predict regulators that might drive the differentiation of SF eTregs and suggested the vitamin-D receptor (VDR) to be a key regulator of eTreg differentiation.

This is an interesting study and albeit being mainly descriptive adds to our understanding of eTreg differentiation in an inflamed/autoimmune environment. The strength of the study is the comparative analysis of Treg transcriptional profiles derived from PB and inflamed tissue (SF) from defined inflammatory/autoimmune diseases. The use of two data sets from different inflammatory/autoimmune diseases (JIA and RA) is appreciated. The manuscript is well written, the study is well designed and the figures are excellent.

We would like to thank the reviewer for her/his time and positive feedback. We are pleased to read that the reviewer finds the paper to be interesting, and are happy to address the questions that were raised.

I have the following questions/comments:

-The manuscript could benefit from functional validation, especially of the “predicted key-regulator in eTreg differentiation” VDR. Did the authors try to assess the in vitro impact of vitamin D on the induction of the eTreg expression signature?

We have now performed additional in vitro experiments demonstrating that vitamin D can indeed promote expression of core (e)Treg genes and proteins including IL2RA, CTLA4, IL10, CD30 (TNFRSF8), VDR, TBX21 and IRF4 in human Treg (see line 245-254 and Figure 5e and 5f).

- References are missing for mentioned literature (p.6, line 126 „Based on recent literature ...“ and p.6 line 139 „... were previously associated with an eTreg profile ...”)

We thank the reviewer for the alertness and have added the proper references.

- Figure 3:d The differences in expression of the analyzed proteins (T-bet, CXCR3, IL12RB2) and particularly the expressed cytokines IFN- γ and IL-2 at least in non-Tregs might be biased by a higher frequency of memory T cells within the SF compartment that are per se enriched in e.g. cytokine expressing/polarized cells

We recognize that the higher frequency of memory T cells within the SF compartment has an impact on the expression of the analyzed proteins. However, also when solely assessing the CD45RO⁺ cells the increase in expression of those markers in the Treg and non-Treg remain. We have included these new analyses in the manuscript, see Supplementary Fig. 2d, 3c and 4e as well as line 166-167.

- Figure 4: The authors should comment why PB Treg from healthy adults were used for epigenetic analysis whereas PB Treg from children (controls and JIA) were used for transcriptional analysis

We agree that addition of PB Treg from children would be a nice addition for the epigenetic analyses. However, 500.000-1.000.000 cells were required for these experiments and unfortunately the volume of blood allowed to be withdrawn from these children, and consequently the number of Treg that can be isolated and sorted, is not sufficient for these assays. As our transcriptome analyses (and that of others; Peeters et al Cell Reports 2015) demonstrated that T lymphocytes from the blood of JIA patients are relatively similar to that of adult blood, we do not anticipate that this would have large implications for our study.

- p. 9, line 206: “we found super enhancers associated with markers not previously related to (e)Treg differentiation and mostly specific for human Treg ...” -> quote references

We have now added the appropriate references.

- Materials and Methods: the description of flow cytometric analysis regarding analysis of protein expression in SF Treg is almost completely missing, e.g. antibodies used for analysis of CTLA-4, TIGIT, PD1, ICOS ... and stimulation procedures and analysis of cytokine expression (IL-2 and IFN- γ)

We apologize for not having included all flow cytometry procedures and antibodies used in both the summary report and the method section of the paper. It has now been included in the method section under the subheading ‘Cell phenotyping’ (lines 482-496).

- p.24, line 539: “Vitamin D receptor and 1,25 vitamin D3 incubation assay” -> no incubation assay is described in this paragraph. Has this been performed?

We have now performed these experiments and have included these data in the manuscript (lines 245-254, Fig. 5e and 5f) and adapted the method section as such (lines 586-602).

Reviewer #3 (Remarks to the Author):

I want to thank the authors and editors for the opportunity to review this manuscript by Mijnheer and colleagues. In this paper, the authors show that that FOXP3+ T cells (Tregs cells) isolated from synovial fluids of children with juvenile idiopathic arthritis (and adult RA) show a distinctive epigenetic and expression profile that strongly resemble so-called effector Tregs (eTreg) previously identified in mice. A refreshing aspect to this paper is the fact that the authors recognize that the differences between the Tregs seen in the peripheral blood of children with JIA and those identified in their synovial fluid are likely a reflection of normal homeostatic mechanisms. The pediatric and adult rheumatology literature is, unfortunately, awash in papers where this idea is not considered, and where the authors have concluded that differences between peripheral blood and synovial fluid cells is prima facie evidence of “dysregulation” of the synovial fluid cells. I have a few comments and suggestions that I hope the authors will find helpful.

We would like to thank the reviewer for the helpful feedback and hope the comments are found to be sufficiently addressed.

1. Throughout the paper, the authors assume that H3K4me1/H3K27ac-marked regions are functioning enhancers. While these chromatin marks provide strong evidence that the region of interest has enhancer function, such function must be demonstrated experimentally (eg., Kessler et al, PLoS One 2020). It might be awkward to call the H3K4me1/H3K27ac-marked regions “putative enhancers” or “likely enhancers” everywhere they are referred to in the text, but some acknowledgement early in the discussion that H3k4me1/H3K27ac marks are not prima facie evidence of enhancer function would enhance the scientific rigor of the paper.

We agree with the reviewer and have adapted the manuscript accordingly (line 344-346).

2. I’m not sure at all what the authors mean by “super-enhancer associated genes.” Are they referring to the genes most proximal to the enhancers? This term may be applicable to intronic enhancers, which often regulate the gene in which they are located, and they show such enhancers in Figure 4e. However, in the broader context, enhancers often regulate genes many kB away, and not always the most proximal genes. However, Gasperini et al (Cell 2019) have shown that >70% of enhancers regulate genes within the same chromatin loop of topologically associated domain (TAD). Because we still have an incomplete catalogue of what genes are regulated by what enhancers, it’s hard to know what the authors’ term “enhancer-associated genes” means. This reviewer would suggest that lines 192-214 be re-written.

(Super-)enhancer associated genes were determined based on the nearest TSS to the center of the enhancer and super-enhancer locus using the ROSE algorithm. We agree with the reviewer that these enhancers could regulate multiple genes (in a TAD) or many kb away that are missed by this analysis method. Because the spatial chromatin organization of these cells under these conditions have not been examined we are unable to know which exact regulatory units regulate which genes. We have now clarified our analyses in the main text (line 194-195) and discussed this issue in the discussion section of the manuscript (line 347-350).

3. The points being made in Figure 4e would have been more clear if the authors had also provided the RNAseq track with the ChIPseq tracks.

We have now added RNA-seq tracks for PB child Treg and SF Treg, the comparisons mostly made on transcriptomic level, for both BATF and VDR in Fig. 4e.

4. Line 229 – “...these observations demonstrate that the inflammation-adapted effector phenotype...is regulated at the epigenetic level.” I’m wondering, “As opposed to what?” What was the alternative if not epigenetic re-organization? Indeed, we’re learning that for terminally-differentiated cells, re-organization of enhancers (and therefore epigenetic profiles) is the mechanism through which fine-tuned adaptation to local environments occur (e.g., Gosselin et al, Cell 2014). The key finding in this paper isn’t that the transcriptional and functional differences between PB and SF Tregs are associated with re-organization of H3K4me1/H3K27ac marks. That was entirely predictable. What’s interesting is that these cells share common features with other Tregs in other chronically inflamed tissues.

We fully agree with the reviewer that the wording we does not reflect what’s most interesting here. We have adapted the text accordingly (line 229-232).

5. In that line, it would be intriguing to see whether the CD4/FoxP3+ cells from other

chronically inflamed tissues share similar features to the cells described in this paper. The most accessible for individuals in a large children's hospital might be the CD4+ T cells found in the lungs of patients with cystic fibrosis.

This is an intriguing question. We asked the department specialized in cystic fibrosis (Prof. Jeffrey Beekman, UMC Utrecht) but they observe very few T cells in bronchoalveolar lavages from these patients, which makes it impossible to obtain sufficient numbers of Treg. The GEO database for datasets, to our knowledge, does not contain Treg datasets of non-tumor-infiltrating settings that also include PB Treg as comparison. However, our group recently published a paper about Treg in the uterine environment (Wienke et al., JCI insight 2020). Although the uterus is not a true inflammatory environment, this environment undergoes extensive remodeling with an immunological component (to prevent rejection of the fetus). Here we also see an effector Treg profile characterized by increased protein and mRNA expression of FOXP3, ICOS, GITR, PD-1, TBX21, CXCR3, and on pathway level IL-12 mediated signaling among others, although, somewhat less pronounced than in the SF-derived Treg. This indicates that amongst diverse settings Treg adapt in a similar pattern.

Some minor comments

1. This reviewer was surprised to see the word “men” to describe humans (lines 12 and 342). In my own institution, this would be considered exclusive and a faux pas. I am aware that the senior author, Dr van Wijk, is a talented and respected scientist who is also a woman.

We have adapted the use of ‘men’ to humans.

2. Some of the expansive language was a little off-putting for this reviewer, e.g., “eminently” (p. line 69) “highly relevant” (line 71). My own view is that, if you need an adverb to modify an adjective, you might want to select another adjective. I think this rule works in any Indo-European language.

We have adapted the manuscript accordingly.

Reviewer #4 (Remarks to the Author):

The manuscript by Mijnheer et al. examined transcriptional and epigenetic profiles of the human effector Tregs, and showed the unique transcriptional patterns and similarities to other types of Tregs such as tumor-infiltrating Tregs. Based on the analysis the authors claimed that VDR functions as a key-regulator in effector Treg differentiation; however, simple comparisons only enumerate the possibilities. Conclusions should be drawn through experimental confirmations.

We would like to thank the reviewer for the suggestions and hope they are now sufficiently addressed.

In several cases, the authors claimed the similarities between SF Tregs and selected characteristics via GSEA. GSEA can examine the similarities with cells of your own choosing only. More objective views or a comparison with multiple cells would be required. For example, it has been reported that a variety of Tregs, such as effector Tregs, naïve Tregs, exhausted Tregs, exTregs, tissue-resident Tregs, activated Tregs, tumor-infiltrating Tregs were present in our bodies. Therefore, the authors should examine the similarities of SF Tregs

to these Tregs, and then claim that one specific population shows the highest similarities to SF Tregs.

We have now performed a broad comparison of all immunological signatures collected in database C7 of the Broad Institute that concern Treg (both mouse and human). Since this database is more limited with regards to recent literature we also searched for gene sets derived from primarily human literature regarding Treg signatures. This resulted in a total of 369 gene sets.

As expected, gene sets with genes upregulated in Treg compared to effector T cells are enriched in SF Treg compared to PB child Treg. Furthermore, from this comparison we can conclude that SF Treg show no signs of being naïve nor exhausted or exTregs. However, the other types of Treg are all to some extent enriched in SF Treg compared to PB child Treg. This includes tissue-resident Treg (e.g. lung, colon, fat, liver, and shared tissue gene sets), activated Treg (e.g. IL-2, CD3/CD28 stimulated), and tumor-infiltrating Tregs (e.g. colon, breast, liver, skin).

When performing a leading edge analysis on those gene sets enriched in SF Treg the core eTreg genes that we have identified both on the transcriptome and epigenome level, including, BATF, CCND2, CCR8, TNFRSF18, and VDR, are shared with the presented data-sets. It thus appears that upon activation of Treg to become effector Treg, and potentially migrate to a tissue, site of inflammation or tumor environment, a shared set of core-genes is upregulated for all these settings.

To claim one specific population shows the highest similarities to SF Tregs is hard without a direct comparison using the exact same methodological approach. Amongst others, the species of origin, depth of sequencing, mapping, cut-off used for differential gene expression can all influence the set of genes defining a Treg subtype as a result. We now discuss this overlap in the last results paragraph (line 294-298), and have included this GSEA as well as the leading edge analysis as Supplementary Table 4.

The authors claimed that VDR functions as a key-regulator in effector Treg differentiation. Does VDR repression (for example, via Crisper-Cas9, siRNA, KD) interfere the differentiation of effector Tregs? Conversely, does Vitamin D enhance the differentiation? Such experimental confirmation would be essential.

We have now performed additional in vitro experiments demonstrating that vitamin D can indeed promote expression of core (e)Treg genes and proteins including IL2RA, CTLA4, IL10, CD30 (TNFRSF8), VDR, TBX21 and IRF4 in human Treg (see line 244-254 and Figure 5e and 5f).

The authors also claimed that autoimmune inflammation-derived Tregs display a unique transcriptional profile. However, FOXP3, CTLA-4, TIGIT, GITR, etc. would be upregulated in Tregs by TCR stimulation (CD3/CD28 antibody stimulation in vitro), and also tissue-resident Tregs express BATF. TCR stimulation-dependent transcriptional changes and changes in tissue-resident cells would be included in the comparison.

Indeed, some of the genes discussed in our paper are upregulated by Tregs that have received TCR stimulation, and some genes are also upregulated upon becoming a tissue-resident Treg. Comparisons to these and other signatures have now been included in Supplementary Table 4 by a gene set enrichment analysis of Treg subtype gene sets including TCR stimulation and tissue residence.

Our findings concern a distinctive profile from PB Treg, with commonalities amongst the defined Treg subtypes, so we agree 'unique' is not the most suitable wording and have removed it.

Minor comments:

In Fig. 1b, control samples are required in the Treg suppression assay.

We have now added both the negative (PBMC without CD3/CD28 stimulation and Treg) and positive (PBMC with CD3/CD28 stimulation but without Treg) control to Fig. 1b.

This reviewer could not understand the graphical details and an importance of Fig. 5a. Improvement would be required.

The aim with Fig. 5a is to show that there is a set of key regulators, , involved in PB to SF Treg differentiation, and that most regulators are interconnected. We have rewritten the figure legend regarding Fig. 5a to more clearly explain what is shown, and added extra clarification in the main text (line 235 and 237).

REVIEWERS' COMMENTS

Reviewer #1 (Remarks to the Author):

The authors have significantly modified and improved the manuscript. All changes are clearly indicated. I have only one minor comment: on page 20, in the methods section, lines 442-443, the sentence can be shortened by removing "clinical" and "clinically" so that the phrases are simply "active disease" and "inactive disease."

Reviewer #2 (Remarks to the Author):

The reviewer would like to thank the authors for addressing the comments. The authors have addressed all the comments of the reviewers and revised the manuscript accordingly. Specifically, they have added functional data addressing the promotion of the (e)Treg core program by vitamin D.

From my point of view, the paper is suitable for publication in Nature Communications.

Please address a very minor point:

references to figures in line 167: it should be "2d" instead of "2c"; also "e" and "f" are mistaken in Figure 4 and the corresponding figure legend.

Reviewer #3 (Remarks to the Author):

I thank the authors for giving me the chance to get another look at this paper and apologize for the delay in my response. The authors have answered all my concerns satisfactorily.

Reviewer #4 (Remarks to the Author):

The manuscript was sufficiently revised for my claims.

REVIEWERS' COMMENTS

Reviewer #1 (Remarks to the Author):

The authors have significantly modified and improved the manuscript. All changes are clearly indicated. I have only one minor comment: on page 20, in the methods section, lines 442-443, the sentence can be shortened by removing "clinical" and "clinically" so that the phrases are simply "active disease" and "inactive disease."

We have adjusted this.

Reviewer #2 (Remarks to the Author):

The reviewer would like to thank the authors for addressing the comments. The authors have addressed all the comments of the reviewers and revised the manuscript accordingly. Specifically, they have added functional data addressing the promotion of the (e)Treg core program by vitamin D.

From my point of view, the paper is suitable for publication in Nature Communications.

Please address a very minor point:

references to figures in line 167: it should be "2d" instead of "2c"; also "e" and "f" are mistaken in Figure 4 and the corresponding figure legend.

We have adjusted this

Reviewer #3 (Remarks to the Author):

I thank the authors for giving me the chance to get another look at this paper and apologize for the delay in my response. The authors have answered all my concerns satisfactorily.

Reviewer #4 (Remarks to the Author):

The manuscript was sufficiently revised for my claims.